# `CellBRIDGE`: Learning Cellular Trajectories via Interaction-Aware Alignment

Silas Ruhrberg Estévez [* 1]   Nicolas Huynh [* 1]   Tennison Liu [1]   Roderik M. Kortlever [2]   Gerard I. Evan [2]
David L. Bentley [3]   Mihaela van der Schaar [1]

## Abstract

Inferring dynamics from population snapshots is a fundamental challenge in machine learning and biology. In scRNA-sequencing (scRNA-seq), destructive measurements preclude direct tracking of individual cells across time, making trajectory inference underdetermined. Optimal Transport (OT) provides a principled framework for snapshot *alignment*, but a long-standing modeling question is which *cost functions* yield biologically meaningful couplings. Standard OT approaches rely on gene-expression distances, implicitly treating cells as independent points and neglecting structured cell–cell communication mediated by ligand–receptor signaling. We introduce `CellBRIDGE` (*Cell-Based Regularized Interaction-Driven Gene Expression*), which augments feature-based OT with a directed, typed interaction cost derived from ligand–receptor activity. By explicitly modeling cell–cell communication, `CellBRIDGE` improves cross-snapshot couplings and downstream trajectory estimates across synthetic and real scRNA-seq datasets relative to feature-only baselines. Notably, `CellBRIDGE` enables mechanistically interpretable *in silico* perturbations: on lung cancer data, silencing specific ligand–receptor pairs induces trajectory shifts that recapitulate expected effects of targeted pathway inhibition.

## 1. Introduction

Understanding how cellular populations evolve over time is fundamental to development, disease, and therapeutic intervention (Yeo et al., 2022; Qiu et al., 2022). Single-cell RNA sequencing (scRNA-seq) measures gene expression at un-precedented resolution, but its destructive nature precludes tracking individual cells across time, making trajectory inference from population snapshots inherently underdetermined (Schiebinger et al., 2019; Bunne et al., 2024). The ability to infer the trajectories of single cells has major implications for drug discovery, where experiments to probe mechanisms and interventions are costly and slow (Sertkaya et al., 2024): *in silico* dynamics can guide experiment design and prioritize targets (Yue & Dutta, 2022).

**Challenges of inferring cellular dynamics.** Learning the trajectories of individual cells, i.e. the task of *trajectory inference* (Bunne et al., 2024), requires reconstructing smooth dynamics from unaligned snapshots. This presents a unique challenge: because measurements are destructive, the same cell cannot be observed at multiple time points. These difficulties are further exacerbated by imbalanced cell populations and the noisy, sparse nature of gene expression (Adil et al., 2021; Schiebinger et al., 2019).

**From graph heuristics to couplings.** Classical approaches build a cell–cell $k$NN graph and extract pseudotime and branches via diffusion distances or spanning-tree heuristics (Haghverdi et al., 2016; Street et al., 2018). These locality-based methods assume that proximity within a snapshot reflects temporal adjacency, which can yield biased pseudotimes and spurious lineage structure (Weiler et al., 2022). To address these limitations, more recent methods recast alignment as the task of finding a coupling between *distributions*.

**Biologically meaningful cost functions.** A popular approach for distributional alignment is Optimal Transport (OT) (Peyré & Cuturi, 2019). While OT makes the search for a coupling computationally tractable, the biological validity of the result hinges entirely on the choice of a cost function. As noted by Bunne et al. (2024), incorporating meaningful priors via this cost is a *central bottleneck* in single-cell and spatial omics. Standard OT approaches rely on gene-expression distances, effectively enforcing a principle of least action, assuming that cells evolve smoothly via the shortest path in expression space.

 **In this work, we ask:** *Can we design a biologically meaningful prior for trajectory inference, which is orthogonal to the principle of least action in gene expression?*

---
[*]Equal contribution   [1]DAMTP, University of Cambridge [2]Francis Crick Institute [3]University of Colorado Anschutz Medical Campus.   Correspondence to: Silas Ruhrberg Estévez <sr933@cam.ac.uk>, Nicolas Huynh <nvth2@cam.ac.uk>.

*Proceedings of the 43rd International Conference on Machine Learning*, Seoul, South Korea. PMLR 306, 2026. Copyright 2026 by the author(s).

We begin with a key observation: feature-only OT, which relies solely on gene expression distances, implicitly treats cells as independent particles. This discards structured *cell–cell interactions* (CCIs) and ignores the biological reality that trajectories are shaped by intercellular signaling. Specifically, directed CCIs mediated by ligand–receptor (LR) pairs drive development and disease (He & Xu, 2020; Liu et al., 2023). We posit that the *relational structure* of these interactions can also evolve smoothly over time, and hence can offer a robust signal for alignment.

We incorporate this prior via `CellBRIDGE` (*Cell-Based Regularized Interaction-Driven Gene Expression*). To avoid reliance on spatial data, we construct *proxy* communication networks within each snapshot by scoring directed LR pairs across local expression neighborhoods. We then frame the search of a coupling as a *Fused Gromov–Wasserstein* (FGW) problem. FGW simultaneously minimizes the cost of transport in gene expression space and the structural distortion of these inferred communication networks.

Importantly, our interaction-aware prior is orthogonal to standard priors (such as least action in gene expressions or unbalanced transport). As a consequence, this modularity enables `CellBRIDGE` to be seamlessly integrated into state-of-the-art pipelines for velocity field regression (Lipman et al., 2024; Kapusniak et al., 2024; Tong et al., 2024b). In our experiments across synthetic and real-world datasets, we demonstrate that `CellBRIDGE` leads to improved trajectory inference, with best results obtained when paired with orthogonal priors. To summarize, our contributions are the following:

---

**Contributions**

**Cost-function view.** We identify OT cost design as a key design choice for snapshot alignment, and propose the smoothness of directed, typed cell–cell communication as a biologically grounded and interpretable prior, which complements gene expression smoothness.
**Typed, directed FGW.** We generalize FGW to *multi-relation* (vector-valued) directed interaction structure derived from ligand–receptor signaling, yielding interaction-aware couplings between snapshots.
**Broad applicability.** Interaction-aware couplings improve trajectory inference across various trajectory inference frameworks, showing that CCI structure is a general-purpose prior orthogonal to conventional assumptions.
**Empirical and mechanistic evidence.** We demonstrate improved performance on synthetic and real scRNA-seq datasets, and show interpretable *in silico* perturbations: silencing specific LR pairs induces trajectory shifts consistent with targeted pathway inhibition.

---

## 2. Background

**Problem formulation: cell trajectory inference.** We consider $k$ population snapshots $\{\mathcal{D}_i\}_{i=1}^k$, where each $\mathcal{D}_i \subset \mathbb{R}^d$ is a set of single-cell states measured at time $t_i$. The goal is to learn a time-continuous flow $\psi : \mathbb{R}^d \times \mathbb{R}_+ \to \mathbb{R}^d$ such that $\psi(x, t)$ returns the state obtained by evolving an initial state $x$ to time $t$. Because scRNA-seq is *destructive*, the same cell cannot be observed at two times, so there is no one-to-one correspondence between cells in $\mathcal{D}_i$ and $\mathcal{D}_{i+1}$. Classical time-series and ODE-fitting methods that require repeated observations of the same object are thus not directly applicable; trajectory inference must instead recover dynamics from *unaligned snapshots*.

**Global alignment of snapshots.** Rather than inferring trajectories from neighborhoods within a single snapshot (Haghverdi et al., 2016), recent work aligns *multiple snapshots at the population level* (Schiebinger et al., 2019), treating each snapshot as a probability distribution over cell states. This alignment is inherently underdetermined: without additional structure, many matchings between snapshots are equally compatible with the observed marginals.

**Standard OT for snapshots.** For two timepoints $t_0 < t_1$ with datasets $\mathcal{D}_0 = \{x_i\}_{i=1}^{n_0}$ and $\mathcal{D}_1 = \{y_j\}_{j=1}^{n_1}$, where $x_i, y_j \in \mathbb{R}^d$ are gene-expression vectors, we form the empirical measures $\rho_0 = \sum_{i=1}^{n_0} a_i \, \delta_{x_i}$ and $\rho_1 = \sum_{j=1}^{n_1} b_j \, \delta_{y_j}$, with $a \in \Sigma_{n_0}$, $b \in \Sigma_{n_1}$, and $\Sigma_n := \{w \in \mathbb{R}_+^n : \sum_{k=1}^n w_k = 1\}$ (e.g., $a_i = 1/n_0$ for uniform weights). The alignment problem seeks a coupling $\Gamma^\star \in \Pi(a, b)$ between $\rho_0$ and $\rho_1$ that respects the marginals $a$ and $b$:

$$\Gamma^\star := \left\{ \Gamma \in \mathbb{R}_+^{n_0 \times n_1} \mid \Gamma \mathbf{1}_{n_1} = a, \ \Gamma^\top \mathbf{1}_{n_0} = b \right\}. \quad (1)$$

**OT as a regularization principle.** Among all couplings in $\Pi(a, b)$, how do we select biologically plausible ones? OT answers this via an optimization problem where the cost function is based on the *least-action prior*: cellular states should evolve smoothly over time, so matchings that incur small feature-wise changes are more likely (Villani et al., 2008; Bunne et al., 2024). Given a cost matrix $C \in \mathbb{R}_+^{n_0 \times n_1}$, where $C_{ij} = c(x_i, y_j)$ is the cost of transporting a unit of mass from $x_i$ to $y_j$, the discrete Kantorovich formulation solves

$$\Gamma^\star \in \arg \min_{\Gamma \in \Pi(a,b)} \langle \Gamma, C \rangle_F, \quad (2)$$

where $\langle \cdot, \cdot \rangle_F$ denotes the Frobenius inner product. The optimizer $\Gamma^\star$ is a *soft* alignment that minimizes expected transport cost under $c(\cdot, \cdot)$. However, because $C$ depends only on expression features, feature-only OT cannot exploit intra-snapshot structure such as cell–cell interactions (CCIs) that coordinate population dynamics.

**Incorporating intra-snapshot structure.** Beyond inter-snapshot feature distances, it is common to have access

*Table 1.* **Related work.** A comparison of inductive biases for aligning single-cell population snapshots.

| Method family | Intuition | Signal used for alignment | How `CellBRIDGE` differs |
|---|---|---|---|
| **Feature-based TI / OT** | Development is assumed to be smooth in gene-expression space. | Expression similarity, pseudotime graphs, or least-action OT costs. | Uses intercellular communication structure rather than only cell-intrinsic transcriptional similarity. |
| **Velocity / dynamic priors** | Transitions should follow a forward-time direction and may include growth or death. | RNA velocity, fate probabilities, proliferation, apoptosis, or unbalanced dynamics. | Regularizes the coupling with directed, typed CCI rather than only temporal directionality. |
| **Spatial / geometric OT** | Alignment should preserve physical proximity or relational geometry. | Spatial coordinates, neighborhood graphs, GW/FGW structural costs. | Does not require spatial measurements and uses ligand–receptor channels rather than generic geometry. |
| **Communication-aware models** | Cell transitions may depend on signals exchanged with other cells. | Learned interaction effects or ligand–receptor-derived features. | Uses interpretable, directed, typed CCI as a plug-and-play OT regularizer. |

to structural information within each snapshot (e.g., communication, adjacency, or interaction motifs). The Kantorovich objective in Equation (2) cannot exploit such information, since it depends only on cross-snapshot costs $C$. The Gromov–Wasserstein (GW) problem extends OT by comparing distributions through their *pairwise relational* structure, favoring couplings that approximately preserve within-snapshot relations across time. This can be viewed as a *structure-stationarity prior*: over the relevant time window, key relational patterns are assumed to evolve smoothly to constrain the otherwise underdetermined alignment. Intuitively, if cell populations maintain consistent communication motifs across snapshots, then cells participating in similar interaction patterns at $t_0$ should map to cells with similar patterns at $t_1$.

We represent intra-snapshot structure by relational matrices $G^{(0)} \in \mathbb{R}^{n_0 \times n_0}$ (source) and $G^{(1)} \in \mathbb{R}^{n_1 \times n_1}$ (target). GW then seeks a coupling $\Gamma^\star \in \Pi(a, b)$ that minimizes the distortion between $G^{(0)}$ and $G^{(1)}$:

$$\min_{\Gamma \in \Pi(a,b)} \sum_{i,k=1}^{n_0} \sum_{j,l=1}^{n_1} L(G_{ik}^{(0)}, G_{jl}^{(1)}) \Gamma_{ij} \Gamma_{kl} \qquad (3)$$

Here $L$ denotes a pairwise distortion function. We treat $G^{(0)}$ and $G^{(1)}$ as generic relational matrices (not necessarily symmetric), with the choice of $L$ determining the notion of structure preservation. Finally, the Fused Gromov–Wasserstein (FGW) problem combines feature matching with structure preservation via a parameter $\alpha \in [0, 1]$:

$$\min_{\Gamma \in \Pi(a,b)} (1-\alpha)\langle \Gamma, C \rangle_F + \alpha \sum_{i,k=1}^{n_0} \sum_{j,l=1}^{n_1} L(G_{ik}^{(0)}, G_{jl}^{(1)}) \Gamma_{ij} \Gamma_{kl}$$
$$(4)$$

Setting $\alpha = 0$ recovers feature-only OT, while $\alpha = 1$ recovers GW. In our setting, $G^{(0)}$ and $G^{(1)}$ are not generic neighborhood graphs but directed, typed ligand–receptor communication networks: mechanistically interpretable structure

obtainable from snapshot scRNA-seq using curated LR catalogs, and editable for counterfactual analyses.

## 3. Related Work

Inferring cellular trajectories from population snapshots is inherently underdetermined: many couplings between consecutive timepoints may be consistent with the observed marginal distributions. Existing methods address this ambiguity by imposing different *inductive biases* on the alignment problem. Table 1 summarizes these biases and contrasts them with our communication-aware regularizer. We refer to Section A for a more detailed discussion, and provide a short summary in what follows.

Classical trajectory-inference methods infer developmental structure from transcriptomic neighborhoods, using pseudotime, branching, or graph abstractions to order cells along putative progressions (Qiu et al., 2017; Wolf et al., 2019). Optimal-transport approaches (Peyré & Cuturi, 2019) provide a population-level alternative by coupling distributions across timepoints under a least-action principle in gene-expression space, as in Waddington-OT and related continuous-time formulations (Schiebinger et al., 2019; Tong et al., 2020). These methods are effective when transcriptional change is smooth, but their alignment signal is primarily cell-intrinsic.

A complementary line of work adds temporal directionality or population-level constraints. RNA velocity and its extensions use spliced and unspliced counts to infer local directions of motion, which can then be combined with transcriptomic similarity to estimate fate probabilities (La Manno et al., 2018; Bergen et al., 2020; Lange et al., 2022). Other approaches relax mass conservation or explicitly model growth and death, thereby accounting for changes in population size across time. Such dynamic priors help orient trajectories, but they typically constrain transitions

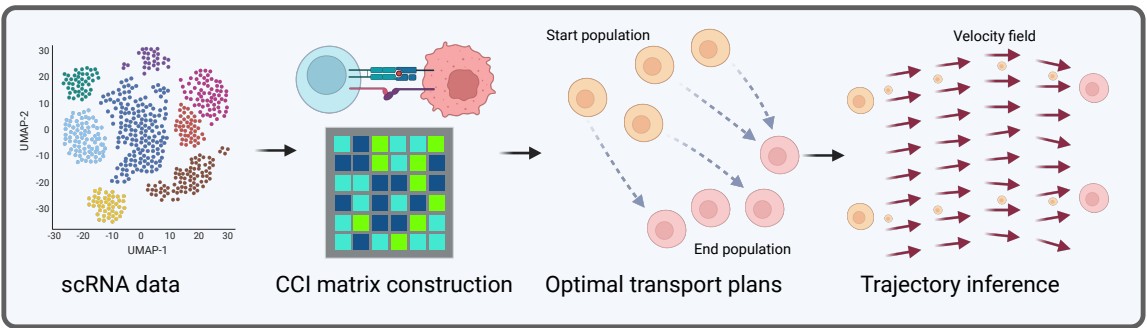

*Figure 1.* **Overview of `CellBRIDGE`.** From an LR catalogue, we build directed CCI matrices. A structure-aware OT problem balances feature similarity with interaction structure to produce snapshot couplings used to train a vector field to recover cell trajectories.

through cell-intrinsic dynamics rather than through structured interactions between cells.

Spatial and geometric methods instead regularize alignment using relational structure. Spatial OT methods exploit measured tissue coordinates to relate cells across modalities or timepoints (Cang & Nie, 2020). These approaches show that relational information can be a powerful alignment signal. However, spatial coordinates are unavailable in standard dissociated scRNA-seq, and generic neighborhood graphs do not specify which cells signal to which others, nor through which ligand–receptor channels.

Our work introduces a complementary communication-based inductive bias. From ligand–receptor expression, we construct a directed and typed representation of cell–cell communication and use it to regularize the OT coupling. Thus, unlike feature-based, velocity-based, or spatial/geometric priors, our method encourages temporal alignments that preserve smoothly evolving signaling roles across time. Related communication-aware models also recognize that cell transitions may depend on intercellular effects (Zhang et al., 2025). However, our contribution is to inject interpretable ligand–receptor communication structure directly at the coupling level, making the regularizer plug-and-play with downstream trajectory learners. More generally, our work integrates information from biological knowledge, an approach followed for other tasks (e.g. cell annotation (Wang et al., 2021; Tang et al., 2024; di Montesano et al., 2026), or gene regulatory networks (Hossain et al., 2024)).

## 4. `CellBRIDGE`: Interaction-Aware Optimal Transport

**Overview.** We study whether incorporating a structural prior on cell–cell interactions (CCIs), i.e. favoring couplings that encode smoothly evolving communication structure across snapshots, improves trajectory inference. We introduce `CellBRIDGE`, which integrates gene-expression features and interaction networks into a unified OT objective. Given

source and target snapshots $\mathcal{D}_0$ and $\mathcal{D}_1$, `CellBRIDGE` first computes a cross-snapshot coupling that assigns source to target cells probabilistically. The coupling is designed to satisfy two desiderata: **(D1) feature coherence**, preserving smooth evolution in expression space, and **(D2) communication evolution smoothness**, favoring couplings that encode a smooth evolution of the directed CCI geometry induced by ligand–receptor expression. We achieve this by solving a Fused Gromov–Wasserstein (FGW) problem that balances feature similarity and CCI structure, yielding $\Gamma^\star$. Because the interaction prior is encoded at the coupling level, $\Gamma^\star$ can be reused as a plug-and-play input to downstream continuous-time models, including flow matching, diffusion Schrödinger bridges (DSB), and unbalanced extensions.

### 4.1. Interaction-aware transport via multi LR-pair FGW

**Modeling cell–cell interactions from scRNA.** Given a ligand–receptor (LR) catalog $\mathcal{P} = \{(l_p, r_p) \mid p \in [P]\}$ of $P$ ligand–receptor pairs and a dataset of $n$ cells, our aim is to construct a directed, nonnegative CCI tensor $G \in \mathbb{R}_{\geq 0}^{n \times n \times P}$ that summarizes potential signaling from any sender cell $i$ to receiver cell $j$. Starting from raw expression counts, we first apply library-size normalization, i.e. rescaling each cell's counts by its total count and multiplying by a fixed scale factor, so that ligand and receptor expression levels are comparable across cells. Rather than a $\log(1+\cdot)$ transform, which compresses high-expression signals logarithmically and can obscure fold-change differences at high abundances, we map each gene to $[0, 1]$ using a Hill saturation function. The Hill form captures saturation/occupancy effects common in receptor systems and yields bounded activations while preserving rank ordering. For gene $g$ and cell $c$, we define $s_{cg} = (x_{cg}^{h_g})/(x_{cg}^{h_g} + \kappa_g^{h_g})$, with robust scale $\kappa_g$ (e.g., the $q{=}0.9$ quantile of nonzero values in $\{x_{cg} \mid c \in [n]\}$) and exponent $h_g$ controlling sharpness (we use $h_g = 1$ in our experiments).

This gives bounded activations where near-saturating expression contributes strongly. For an LR pair $p = (l_p, r_p)$ and cells $i$ (sender) and $j$ (receiver), we score the interaction as $z_{i \to j}^{(p)} = s_{il_p} s_{jr_p}$, capturing the intuitive requirement that ligand availability and receptor readiness must co-occur. We then set $G_{ijp} = z_{i \to j}^{(p)}$. The CCI tensor $G$ serves as the directed, multi-channel structure we aim to preserve during cross-snapshot alignment [1].

**Interaction-aware transport via multi LR-pair FGW.** Given two snapshots $\mathcal{D}_0 = \{x_i\}_{i=1}^{n_0}$ and $\mathcal{D}_1 = \{y_j\}_{j=1}^{n_1}$, we define a feature cost matrix $C \in \mathbb{R}_{\geq 0}^{n_0 \times n_1}$ such that $C_{ij} = c(x_i, y_j)$ (typically squared Euclidean distance). From the CCI construction above, we obtain directed, non-negative tensors $G^{(0)} \in \mathbb{R}_{\geq 0}^{n_0 \times n_0 \times P}$ and $G^{(1)} \in \mathbb{R}_{\geq 0}^{n_1 \times n_1 \times P}$ for the source and target snapshots. Our objective is to find a coupling $\Gamma \in \mathbb{R}_{\geq 0}^{n_0 \times n_1}$ that aligns cells while respecting CCI structure. Let $g_{i \to k}^{(0)} := G_{ik:}^{(0)} \in \mathbb{R}_{\geq 0}^{P}$ and $g_{j \to \ell}^{(1)} := G_{j\ell:}^{(1)} \in \mathbb{R}_{\geq 0}^{P}$ denote LR-channel vectors for directed edges. We seek a coupling that is a solution to the following optimization problem:

$$
\begin{aligned}
\min_{\Gamma \in \Pi(a,b)} \quad & (1-\alpha)\,\mathcal{F}(\Gamma) + \alpha\,\mathcal{S}(\Gamma), \\
\mathcal{F}(\Gamma) &= \langle \Gamma, C \rangle, \\
\mathcal{S}(\Gamma) &= \sum_{i,k=1}^{n_0} \sum_{j,\ell=1}^{n_1} \varphi\big(g_{i \to k}^{(0)}, g_{j \to \ell}^{(1)}\big) \Gamma_{ij} \Gamma_{k\ell}.
\end{aligned}
\tag{5}
$$

The similarity $\varphi(\cdot, \cdot)$ measures how well the *interaction profile* between a sender–receiver pair in the source snapshot matches that of a pair in the target snapshot. Intuitively, we penalize couplings that map cells in one snapshot to cells in the other snapshot when doing so would mismatch their directed, multi-channel signaling context. By default, we use squared Euclidean distance $\varphi(u, v) = \|u - v\|_2^2$, which is natural here because interaction vectors are bounded in $[0,1]^P$ and the loss decomposes across LR channels. Unlike the classical FGW setting (Vayer et al., 2020), CellBRIDGE operates on *multi-typed* interactions, since each directed relation is a vector in $\mathbb{R}^P$ rather than a scalar.

**Optimization.** Equation (5) is non-convex due to the quadratic structure term (note that we do not use entropic regularization in our objective because it would favor more diffuse couplings). We solve it using a customized conditional-gradient routine adapted from Braun et al. (2022), detailed in Section D.3. Because the feature costs $C$ and interaction tensors $G^{(m)}$ can have different magnitudes and units, the trade-off parameter $\alpha$ in Equation (5) does not, by itself, guarantee a meaningful balance. Without

---

[1] The preprocessing described here is used only for constructing the CCI tensors. Gene-expression features used for the OT feature cost, downstream models, and baselines follow the standard preprocessing pipeline described in Section D.1.

normalization, one term can dominate the objective, making $\alpha$ difficult to interpret and tune. To balance the two contributions, we normalize by *endpoints*: we solve the feature-only problem ($\alpha = 0$) and the structure-only problem ($\alpha = 1$), obtaining reference objective values $F^\star = \langle \Gamma_{\alpha=0}^\star, C \rangle_F$ and $S^\star = \mathcal{S}(\Gamma_{\alpha=1}^\star)$. We then rescale the objective so that intermediate $\alpha$ values interpolate comparably between these extremes (details in Section D.4).

### 4.2. From couplings to continuous dynamics

**Coupling-level prior.** A key design principle of CellBRIDGE is that interaction structure is injected *only at the level of the cross-snapshot coupling*. Solving the interaction-aware FGW problem in Equation (5) yields a coupling $\Gamma^\star$ between $\mathcal{D}_0$ and $\mathcal{D}_1$. Crucially, $\Gamma^\star$ can be reused by any downstream method that learns continuous-time dynamics from paired endpoints. Thus, CellBRIDGE provides a *plug-and-play structural prior* that is orthogonal to the choice of dynamics model.

**Coupling-induced endpoint distribution.** From the coupling $\Gamma^\star$ we form a joint distribution $\Pi = \sum_{i,j} \bar{\Gamma}_{ij}^\star \delta_{(x_i, y_j)}$, $\bar{\Gamma}_{ij}^\star := \Gamma_{ij}^\star / M$, $M := \sum_{i,j} \Gamma_{ij}^\star$, whose marginals match the empirical snapshot measures $\rho_0$ and $\rho_1$ induced by $\mathcal{D}_0$ and $\mathcal{D}_1$. Downstream continuous-time models differ in how they construct intermediate-time states between paired endpoints. We capture this choice through a (possibly stochastic) *interpolant* $I_t$ (Albergo et al., 2023): $(X, Y) \sim \Pi$, $Z_t = I_t(X, Y, \xi)$, $\rho_t := \text{Law}(Z_t)$, where $\xi$ denotes auxiliary randomness (absent for deterministic interpolants). Intuitively, $\Pi$ fixes *which* endpoints are paired (and with what mass), while the interpolant specifies *how* mass is distributed at intermediate times.

Many trajectory-learning objectives can be written as regressing one or more model fields to targets induced by the chosen interpolant. Let $f_\theta^{(m)}(z, t)$ denote model outputs (e.g. a velocity head and/or a score head), and let $\tau_t^{(m)}(z \mid X, Y, \xi)$ be the corresponding interpolant-induced targets (e.g. conditional velocity, conditional drift, or conditional score). We consider the generic objective

$$
\begin{aligned}
\mathcal{L}(\theta) &= \mathbb{E}_{\Pi, t, \xi}\left[ \sum_{m \in \mathcal{M}} \lambda_m(t) \left\| f_\theta^{(m)}(Z_t, t) - T_t^{(m)} \right\|_{(t)}^2 \right], \\
T_t^{(m)} &= \tau_t^{(m)}(Z_t \mid X, Y, \xi).
\end{aligned}
\tag{6}
$$

where the expectation is over $(X, Y) \sim \Pi$, $t \sim \text{Unif}[0,1]$, $\xi$, and $Z_t = I_t(X, Y, \xi)$. Here, $\lambda_m(t)$ are scalar weights and $\| \cdot \|_{(t)}$ is either Euclidean or metric-induced. CFM (Lipman et al., 2024), SF2M (Tong et al., 2024b), and MFM (Kapusniak et al., 2024) correspond to particular choices of interpolant $I_t$ and targets $\tau_t^{(m)}$, while CellBRIDGE specifies the endpoint distribution $\Pi$.

**Conditional Flow Matching (CFM).** CFM (Lipman et al., 2022) is traditionally implemented with the deterministic affine interpolant $I_t(X, Y) = (1 - t)X + tY$, i.e. $Z_t = (1 - t)X + tY$. With a single velocity head $f_\theta^{(v)} = v_\theta$, the target is constant $\tau_t^{(v)}(Z_t \mid X, Y) = Y - X$, yielding:

$$\mathcal{L}_{\text{CFM}}(\theta) = \mathbb{E}_{\substack{(X,Y)\sim\Pi \\ t\sim\text{Unif}[0,1]}} \left[ \left\| v_\theta(Z_t, t) - (Y - X) \right\|_2^2 \right]. \tag{7}$$

Trajectories are then obtained by integrating $\dot{z}(t) = v_\theta(z(t), t)$.

**Stochastic bridges via SF2M.** SF2M (Tong et al., 2024b) replaces the deterministic affine interpolant by a stochastic interpolant, e.g. sampling $Z_t$ from a Brownian bridge between endpoints $(X, Y) \sim \Pi$, which provides closed-form conditional drifts and scores at intermediate times. We combine `CellBRIDGE` with SF2M by instantiating the endpoint distribution with $\Pi$ (i.e. sampling $(X, Y) \sim \Pi$), while keeping the SF2M training objective and its interpolant-induced targets unchanged. We provide more details in Section E.4.

**Geometric priors via Metric Flow Matching (MFM).** MFM (Kapusniak et al., 2024) corresponds to a geometry-aware interpolant in which intermediate states follow geodesic interpolants under a learned, data-dependent Riemannian metric. In Equation (6), this amounts to choosing $Z_t = I_t^{\text{geo}}(X, Y)$ (or its learned approximation), using a metric-induced norm $\|\cdot\|_{(t)}$, and setting the target to the associated geodesic velocity $\tau_t^{(v)} = \partial_t I_t^{\text{geo}}(X, Y)$. We combine `CellBRIDGE` with MFM by reusing the same endpoint pairs sampled from $\Pi$, while replacing linear interpolation with the learned geodesic interpolant (see Section E.3).

**Unbalanced dynamics.** When total population mass changes between snapshots (e.g. proliferation or apoptosis), we extend `CellBRIDGE` with an unbalanced OT formulation to infer non-uniform marginals prior to solving the interaction-aware FGW problem (see Section E.2). The resulting coupling (and thus $\Pi$) can again be passed unchanged into any downstream choice of interpolant and matching objective, including CFM, SF2M, or MFM.

## 5. Experiments

We evaluate whether incorporating cell–cell interaction (CCI) structure improves cross-snapshot alignment and downstream continuous-time trajectory inference, and whether the resulting inductive bias is biologically grounded. Our evaluation is organized around four questions: **(Q1) Couplings:** does `CellBRIDGE` produce more faithful transport maps than feature-only baselines? **(Q2) Trajectories:** do improved couplings translate into improved continuous-time dynamics? **(Q3) Grounding:** are the gains

driven by biologically meaningful ligand–receptor (LR) structure? **(Q4) Failure modes:** when does the interaction prior become uninformative or harmful?

### 5.1. Do interaction-aware couplings improve cross-snapshot alignment?

Throughout this section, we evaluate the quality of a cross-snapshot coupling by *held-out interpolation*. Given two endpoint snapshots at $t_0 < t_2$, we infer a coupling $\Gamma$ between $\mathcal{D}_0$ and $\mathcal{D}_2$. We then define an intermediate distribution at $t_1$ via affine interpolation along endpoint pairs sampled from the coupling and compare the interpolated marginal to the empirical snapshot at $t_1$. This isolates coupling quality *independently of any downstream dynamics model*. In what follows, the coupling is computed once at the full snapshot level, i.e. using the full source and target snapshots rather than mini-batches [2].

**Synthetic setup.** We consider two 2D snapshots, each composed of three clusters. The second snapshot is obtained by translating each cluster by a distinct vector, inducing a known one-to-one ground-truth transport. We define an interaction structure with $P = 2$ types : the middle cluster points to the left (*Pathway 1*) and to the right (*Pathway 2*), mirrored in the target snapshot (see Section C.1 for more details). We then obtain a coupling for each $\alpha \in \{0, 0.1, \dots, 1\}$ by solving the FGW problem defined in Equation (5) with the ground-truth interaction structures.

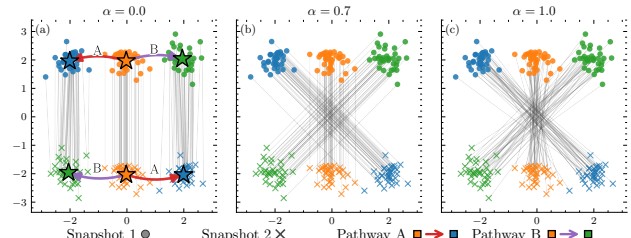

*Figure 2.* **Structure-aware coupling recovers the ground-truth transport map.** We consider within-snapshot interactions with two channels A (orange to blue) and B (orange to green).

**Synthetic results.** Representative couplings across $\alpha$ are shown in Figure 2. With $\alpha = 0$ (feature-only), the interaction structure is ignored and clusters are misaligned; with $\alpha = 1$ (structure-only), interaction types are satisfied but geometry is distorted. An intermediate setting ($\alpha \approx 0.7$) preserves the directed relations while maintaining within-interaction geometry. We refer to Section G.1 for a theoretical analysis of this synthetic setup.

**Real-world datasets.** We evaluate `CellBRIDGE` on six

---

[2]Mini-batching is used only when we train the downstream flow model in Section 4.2, and we sample from the joint distribution induced by the coupling to define the batches.

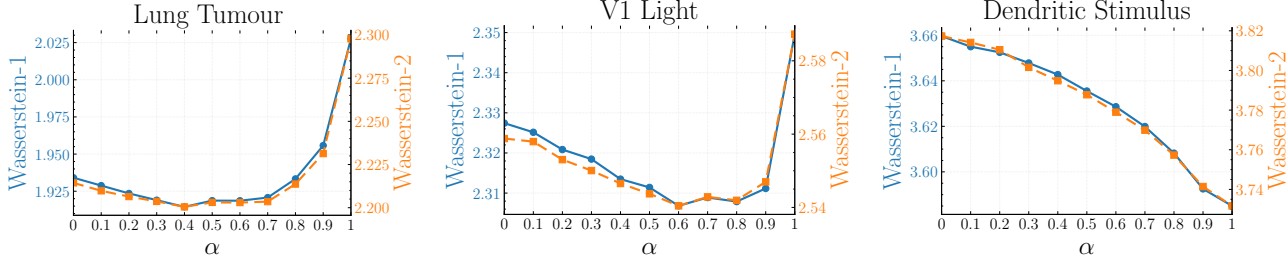

*Figure 3.* **Coupling quality via held-out interpolation.** We plot the $W_1$ and $W_2$ distances between the interpolated and empirical $t_1$ snapshots as $\alpha$ varies. Optimal performance occurs at dataset-specific $\alpha^* > 0$.

*Table 2.* **Interpolation error for continuous-time dynamics (lower is better).** We report mean±std over 5 runs.

| Method | $\alpha$ | V1 Light $W_1$ | V1 Light $W_2$ | Dendritic Stimulus $W_1$ | Dendritic Stimulus $W_2$ | Lung tumor $W_1$ | Lung tumor $W_2$ |
|---|---|---|---|---|---|---|---|
| TrajectoryNet | — | $3.022_{(0.061)}$ | $3.338_{(0.056)}$ | $4.410_{(0.102)}$ | $4.607_{(0.107)}$ | $2.712_{(0.090)}$ | $3.056_{(0.099)}$ |
| DSB | — | $3.819_{(0.152)}$ | $3.875_{(0.143)}$ | $4.099_{(0.155)}$ | $4.249_{(0.153)}$ | $3.700_{(0.116)}$ | $3.967_{(0.102)}$ |
| VGFM | — | $6.446_{(0.114)}$ | $6.745_{(0.102)}$ | $7.087_{(0.022)}$ | $7.261_{(0.026)}$ | $2.175_{(0.017)}$ | $2.478_{(0.019)}$ |
| MIOFlow | — | $6.360_{(0.010)}$ | $6.655_{(0.009)}$ | $6.970_{(0.043)}$ | $7.159_{(0.034)}$ | $2.001_{(0.003)}$ | $2.316_{(0.009)}$ |
| SnapMMD | — | $2.420_{(0.005)}$ | $2.657_{(0.005)}$ | $3.863_{(0.036)}$ | $4.022_{(0.048)}$ | $2.237_{(0.143)}$ | $2.520_{(0.115)}$ |
| Moscot | — | $6.242_{(0.000)}$ | $6.545_{(0.000)}$ | $7.115_{(0.000)}$ | $7.331_{(0.000)}$ | $2.000_{(0.000)}$ | $2.335_{(0.000)}$ |
| OT-CFM | — | $2.392_{(0.005)}$ | $2.625_{(0.007)}$ | $3.696_{(0.007)}$ | $3.857_{(0.009)}$ | $1.993_{(0.004)}$ | $2.275_{(0.005)}$ |
| OT-MFM | — | $2.401_{(0.003)}$ | $2.636_{(0.003)}$ | $3.714_{(0.008)}$ | $3.880_{(0.009)}$ | $1.984_{(0.004)}$ | $2.285_{(0.004)}$ |
| UOT-FM | — | $2.411_{(0.005)}$ | $2.649_{(0.006)}$ | $3.701_{(0.006)}$ | $3.867_{(0.007)}$ | $1.998_{(0.004)}$ | $2.348_{(0.004)}$ |
| SF2M | — | $3.254_{(0.192)}$ | $3.368_{(0.182)}$ | $4.333_{(0.279)}$ | $4.436_{(0.282)}$ | $3.826_{(0.265)}$ | $3.974_{(0.308)}$ |
| CellBRIDGE +SF2M | 0.5 | $3.199_{(0.117)}$ | $3.315_{(0.110)}$ | $4.303_{(0.213)}$ | $4.397_{(0.205)}$ | $3.809_{(0.302)}$ | $3.968_{(0.374)}$ |
| | 1 | $3.226_{(0.075)}$ | $3.339_{(0.073)}$ | $4.289_{(0.110)}$ | $4.387_{(0.108)}$ | $3.638_{(0.308)}$ | $3.739_{(0.335)}$ |
| CellBRIDGE +MFM | 0.5 | $2.393_{(0.007)}$ | $2.631_{(0.008)}$ | $3.679_{(0.007)}$ | $3.838_{(0.009)}$ | $1.978_{(0.004)}$ | $2.277_{(0.003)}$ |
| | 1 | $2.363_{(0.002)}$ | $2.606_{(0.002)}$ | $3.668_{(0.010)}$ | $3.824_{(0.011)}$ | $2.013_{(0.003)}$ | $2.304_{(0.003)}$ |
| CellBRIDGE +UOT-FM | 0.5 | $2.377_{(0.004)}$ | $2.619_{(0.005)}$ | $3.688_{(0.012)}$ | $3.854_{(0.012)}$ | $\mathbf{1.971_{(0.005)}}$ | $2.322_{(0.005)}$ |
| | 1 | $\mathbf{2.360_{(0.002)}}$ | $2.605_{(0.001)}$ | $\mathbf{3.624_{(0.004)}}$ | $\mathbf{3.780_{(0.002)}}$ | $1.993_{(0.004)}$ | $2.335_{(0.005)}$ |
| CellBRIDGE + CFM | 0.5 | $2.381_{(0.004)}$ | $2.618_{(0.003)}$ | $3.679_{(0.009)}$ | $3.835_{(0.010)}$ | $1.989_{(0.004)}$ | $\mathbf{2.272_{(0.005)}}$ |
| | 1 | $2.362_{(0.003)}$ | $\mathbf{2.601_{(0.005)}}$ | $3.639_{(0.021)}$ | $3.788_{(0.021)}$ | $2.057_{(0.005)}$ | $2.329_{(0.005)}$ |

real-world scRNA-seq datasets whose characteristics are summarized in Table 6. We selected these datasets because their temporal coverage provides a favorable window in which ligand–receptor (LR) interactions are expected to remain approximately persistent. Following standard pre-processing, we project gene-expression profiles onto the top $d = 20$ principal components (Section D.1) and standardize them as in Tong et al. (2024a). Additional details on dataset collection are provided in Section C, and results on further datasets are in Section F.

**Setup.** We build CCI tensors by selecting dataset-specific ligand–receptor pairs via an automated procedure that accounts for stability of expression levels across snapshots (cf. Section D.5 for more details). Given three time points $t_0 < t_1 < t_2$, we hold out the snapshot at $t_1$. Using only $t_0$ and $t_2$, and for a chosen LR catalog $\mathcal{P}$ and hyperparameter $\alpha \in \{0, 0.1, \dots, 1.0\}$, we obtain a coupling $\Gamma(\alpha, \mathcal{P})$ by solving the OT problem defined in Equation (5). We define the marginal at $t_1$ by affine interpolation and denote it by $\rho_{t_1}(\alpha, \mathcal{P})$. For each $\alpha$ and $\mathcal{P}$, we compare $\rho_{t_1}(\alpha, \mathcal{P})$ with the empirical distribution $\rho_{t_1}$ observed at $t_1$, computing the Wasserstein-1 and Wasserstein-2 distances.

**Results.** Across datasets, incorporating CCI structure improves alignment, with optimal performance at a dataset-specific $\alpha^* > 0$ (see Figure 3). We observe two regimes: a U-shaped curve with $0 < \alpha^* < 1$, indicating that combining CCI with feature-only OT is best, and an almost monotonic decrease with a minimum at $\alpha^* = 1$ for the *Dendritic Stimulus* dataset. For the latter, this may reflect the dataset's smaller size and coherent stimulation response, which make feature-only OT comparatively less informative than the interaction structure. Taken together with the synthetic results, these findings support (Q1): encoding directed, typed CCI structure improves coupling fidelity in settings where interaction geometry is (approximately) persistent.

### 5.2. Cross-snapshot trajectory inference with learned velocity fields

**Setup.** We evaluate CellBRIDGE against a comprehensive set of state-of-the-art trajectory inference methods to assess whether improvements in coupling quality translate into improved continuous-time dynamics. Starting from a coupling $\Gamma$ inferred between $t_0$ and $t_2$, we learn a time-dependent velocity field and integrate it to transport cells from $t_0$ to

the held-out snapshot at $t_1$. Performance is measured by the Wasserstein-1 and Wasserstein-2 distances between the transported distribution and the empirical distribution at $t_1$. We report results for $\alpha \in \{0, 0.5, 1\}$. We benchmark against representative methods spanning multiple paradigms: neural ODE models (Tong et al., 2020; Huguet et al., 2022), stochastic Schrödinger bridges (De Bortoli et al., 2021; Tong et al., 2024b), flow-matching approaches (Kapusniak et al., 2024; Eyring et al., 2024; Wang et al., 2025), OT baselines (Klein et al., 2025) and kernel-based methods (Berlinghieri et al., 2025). Because performance in continuous-time models can be sensitive to how endpoint correspondences are established, we evaluate `CellBRIDGE` as a *plug-and-play coupling prior*. Specifically, for four representative baselines (`CFM`, `MFM`, `UOT-FM`, and `SF2M`), we replace the default coupling or matching mechanism with the interaction-aware coupling yielding "`CellBRIDGE+`" variants. In these settings, the dynamics model, architecture, and training objective are held fixed, isolating the effect of coupling quality. Implementation details are provided in Sections E.2 to E.4.

**Results.** Table 2 reports interpolation error at the held-out time $t_1$. Across datasets and trajectory-learning paradigms, we observe consistent improvements when replacing feature-only or entropic couplings with the `CellBRIDGE` coupling. Both $\alpha = 0.5$ and $\alpha = 1$ generally improve over the feature-only case ($\alpha = 0$), with the balanced setting $\alpha = 0.5$ providing the most consistent gains across the paired comparisons. A paired Wilcoxon significance analysis confirms that these improvements remain significant after Holm correction, with adjusted $p < 0.05$ for all model families and Cohen's $d_z$ values ranging from 0.27 to 0.63, corresponding to small-to-moderate paired effect sizes. Notably, the same interaction-aware coupling improves multiple downstream methods, demonstrating that gains arise from improved cross-snapshot alignment rather than from method-specific architectural choices.

## 5.3. Are the gains driven by biologically meaningful ligand–receptor structure?

We use `CellBRIDGE` to simulate intercellular perturbations on the *Lung Tumor* dataset by ablating specific pathways from the ligand–receptor catalog, recomputing the CCI tensors, and re-solving Equation (5). This intervention alters only the interaction prior (with baseline expression at $t = 0$ fixed), mimicking a pharmacological blockade prior to transcriptional adaptation (Lee et al., 2016). We quantify trajectory shifts relative to the unperturbed baseline using the 20 Hallmarks of Cancer gene sets (Appendix D.7.1) over a 24h interpolation window.

**Results.** Figure 4a shows the relative decrease in tumour-associated progression scores under different catalog ed-

its. Attenuating signaling through EGFR, ALK, or MET produces measurable reductions (up to 15.5%), indicating that the inferred trajectories are sensitive to these pathways. This aligns with their established therapeutic relevance in non–small cell lung cancer, where EGFR inhibitors (e.g., gefitinib, osimertinib), ALK inhibitors (e.g., crizotinib, alectinib), and MET inhibitors (e.g., capmatinib, tepotinib) are used clinically (Domvri et al., 2013). By contrast, edits to unrelated cardio–renal pathways (RAAS, vasopressin, natriuretic peptides) yield negligible changes, suggesting that `CellBRIDGE` responds specifically to biologically relevant ligand–receptor structure rather than arbitrary perturbations.

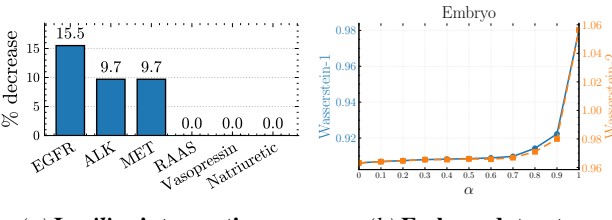

*(a)* **In *silico* interventions.**  *(b)* **Embryo dataset.**

*Figure 4.* **When interaction priors help and when they do not.** Left: pathway-specific LR edits produce measurable trajectory shifts. Right: in rapidly remodeling development, CCI structure may be non-persistent and provides no benefit.

**Ablations.** Motivated by the observation that editing the LR catalog shifts inferred trajectories, we test whether gains are driven by coherent LR structure rather than arbitrary regularization by applying three controlled perturbations to the CCI construction at $\alpha = 1$: *Random LR catalog*—replace the curated LR catalog with a random subset of the same size; *Shuffling*—randomly permute all entries of the CCI tensors, destroying coherent structure; *Metacells*—aggregate cells into metacells before constructing CCIs and then lift interactions back to the cell level (see Section D.2), thereby smoothing the signal. We report the results in Table 3, where we compute the metrics based on the interpolation setup described in Section 5.1. Shuffling the CCI leads to a performance drop, confirming that the *structural organization* of LR interactions drives the gains. Using a random LR catalog also degrades interpolation, highlighting the importance of *LR specificity*. Metacell-based CCI yields intermediate performance by smoothing dropout noise, but can oversmooth, degrading performance, consistent with prior spatiotemporal analyses (Klein et al., 2025).

## 5.4. Does structure always help?

**Setup.** Our formulation does not assume static cell–cell interaction (CCI) structure across time. Instead, as in standard optimal transport, we impose a smoothness principle: among admissible couplings, we favor those that minimize feature displacement while approximately preserving directed interaction geometry. This is appropriate when snap-

*Table 3.* **Sensitivity analysis on the CCI construction.** We report results mimicking the marginal interpolation setup followed in Section 5.1. Baselines are reported as mean $\pm$ standard deviation over five seeds, while CellBRIDGE is deterministic.

| Method | V1 Light | | Dendritic Stimulus | | Lung tumor | |
|---|---|---|---|---|---|---|
| | $W_1$ | $W_2$ | $W_1$ | $W_2$ | $W_1$ | $W_2$ |
| Shuffle | 2.441 (0.002) | 2.646 (0.002) | 3.644 (0.004) | 3.748 (0.007) | 2.188 (0.007) | 2.392 (0.007) |
| Random LR | 2.446 (0.006) | 2.655 (0.008) | 3.602 (0.022) | 3.729 (0.019) | 2.171 (0.030) | 2.388 (0.033) |
| Metacell | 2.329 (0.000) | 2.567 (0.002) | 3.587 (0.005) | 3.725 (0.000) | 2.052 (0.002) | 2.344 (0.002) |
| CellBRIDGE | 2.350 | 2.587 | 3.585 | 3.732 | 2.028 | 2.298 |

shots are separated by modest temporal gaps and the system evolves smoothly. Under rapid, large-scale remodeling this assumption can fail, a known limitation of OT-based alignment rather than a CellBRIDGE-specific issue (Bunne et al., 2023). We illustrate this regime on a developing mouse embryo dataset (Moon et al., 2019), where tissue composition, size, and function shift abruptly and snapshots are six days apart (Qiu et al., 2024).

**Results.** Figure 4b shows that CellBRIDGE provides no improvement over feature-only OT ($\alpha = 0$) on this dataset: interaction structure is not transferable across six-day developmental intervals and becomes uninformative. Accordingly, performance is essentially flat in $\alpha$, with degradation at larger $\alpha$. Practically, when cross-snapshot interaction geometry does not persist, the structural term should be downweighted or omitted.

## 6. Discussion

**Disambiguating alignment via biological structure.** OT–based alignment is often *underdetermined*. CellBRIDGE addresses this bottleneck by injecting a biological inductive bias: a least action principle of ligand–receptor communication. Crucially, LR-derived signaling provides *complementary* information to expression similarity: cells may be transcriptionally similar yet play distinct signaling roles, or conversely exhibit different expression profiles while participating in similar communication patterns. By formulating alignment as a multi-channel FGW objective, CellBRIDGE produces couplings that are simultaneously feature-coherent and consistent with directed, typed communication structure. A practical consequence of making structure an *explicit* prior is editability: because the prior is expressed through the LR catalog used to construct the CCI tensor, CellBRIDGE enables mechanism-specific counterfactuals by quantifying how pathway-level catalog edits shift inferred trajectories.

**A coupling-level, modular prior.** A central design principle of CellBRIDGE is the separation between *structural priors* and *trajectory parameterization*. CellBRIDGE encodes signaling structure once at the level of the cross-snapshot coupling and exposes this as a reusable interface to downstream models. Empirically, we find that

CellBRIDGE improves performance across deterministic flows (CFM), geometry-aware interpolation (MFM), stochastic bridge dynamics (SF2M), and unbalanced transport (UOT-FM), and that replacing baseline couplings with the CellBRIDGE couplings improves those methods. These results suggest that interaction-aware couplings complement advances in generative modeling: they refine the *endpoint correspondence* problem many trajectory learners rely on, without changing architectures or objectives. Finally, negative controls (e.g., random LR assignments or permuted channels) support that gains often depend on biologically meaningful communication structure rather than arbitrary regularization.

**Limitations, extensions, and practical guidance.** CellBRIDGE is most appropriate when cross-snapshot communication structure is at least partially conserved. In rapidly remodeling systems with substantial composition shifts or long temporal gaps, signaling patterns may be non-persistent and the structural term should be downweighted. Several extensions could broaden applicability: (1) *time-varying LR catalogs* to handle non-stationary signaling programs, and (2) tighter *integration with spatial transcriptomics* to validate and refine CCI proxies when spatial coordinates are available. More broadly, scalability to human atlas-scale datasets remains an important direction, both computationally (large-$n$ coupling optimization) and statistically (robust CCI estimation under extreme sparsity).

**Broader impact.** CellBRIDGE provides a general mechanism for population alignment by injecting typed interaction priors into OT. Beyond biology, the same principle applies whenever entities interact through directed, typed relations. For example, in financial networks, directed transaction patterns could regularize market states across regime shifts by constraining correspondences. Analogous ideas apply to social and multi-agent systems, where preserving directed relational structure can reduce ambiguity in cross-time alignment and improve downstream dynamics modeling.

## Acknowledgments

We thank the anonymous ICML reviewers for their comments and suggestions. NH thanks Illumina for their funding and support. TL would like to thank AstraZeneca for their sponsorship and support. The Cambridge Centre for AI in Medicine (CCAIM) receives funding from GSK, Boehringer-Ingelheim, AstraZeneca, Sanofi and Quantum Black, AI by McKinsey.

## Impact Statement

This paper presents work whose goal is to advance the field of Machine Learning. There are many potential societal consequences of our work, none which we feel must be specifically highlighted here. We release the code for `CellBRIDGE` under https://github.com/nicolashuynh/cellbridge and at the wider lab repository https://github.com/vanderschaarlab/cellbridge.

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

# Supplementary Material for `CellBRIDGE`

## A. Extended Related Works

We situate our framework within trajectory inference and optimal transport by organizing prior methods according to the *inductive bias* they impose to resolve the intrinsic underdetermination of aligning population snapshots. Existing approaches regularize alignment through feature smoothness, dynamical directionality, geometric/spatial structure, or conservation laws. Our contribution introduces a complementary bias: smoothness of a *directed, typed cell–cell communication structure*, injected as a plug-and-play regularizer at the level of the OT coupling.

**Feature-based priors: gene-expression smoothness.** Classical trajectory-inference methods reconstruct cellular progressions from neighborhood graphs using pseudotime and branching heuristics (Qiu et al., 2017; Haghverdi et al., 2016; Street et al., 2018; Wolf et al., 2019). When applied to time-course data, these methods typically pool cells from all observed timepoints into a single expression-space graph and then infer pseudotime or branching structure from local transcriptomic neighborhoods. Optimal transport (OT) provides a population-level alternative by coupling entire distributions across timepoints under a least-action bias in gene-expression space (Villani et al., 2008; Peyré & Cuturi, 2019). Waddington-OT (WOT) extends this idea to sequences of snapshots via adjacent-time couplings (Schiebinger et al., 2019), while continuous-time models such as TrajectoryNet learn neural ODE flows constrained by transport (Tong et al., 2020). More recent work connects OT couplings to continuous-time generative dynamics using flow matching or diffusion-based formulations (Klein et al., 2024). These approaches resolve underdetermination by favoring *feature-smooth* matchings, with alignment costs defined in expression space.

**Dynamic priors: directionality and population-level constraints.** A separate line of work encodes *directional* priors or population-scale constraints. RNA velocity and its extensions infer directionality from spliced/unspliced counts and propagate it over $k$NN graphs (La Manno et al., 2018; Bergen et al., 2020), while CellRank combines velocity with transcriptomic similarity to estimate fate probabilities (Lange et al., 2022). Other approaches modify the mass-conservation assumption: unbalanced OT and flow-matching methods such as UOT-FM relax exact mass preservation to account for proliferation or apoptosis (Eyring et al., 2024), while VGFM explicitly models cellular growth rates within generative flows (Wang et al., 2025). Related biologically informed NeuralODE frameworks incorporate mechanistic gene-regulatory priors into continuous-time expression dynamics. For example, PHOENIX (Hossain et al., 2024) uses Hill–Langmuir-inspired neural dynamics and prior GRN structure to learn sparse, interpretable genome-scale regulatory ODEs. However, such models regularize intracellular gene–gene regulatory dynamics rather than the intercellular, directed ligand–receptor communication structure used in our coupling-level prior. Meta Flow Matching (Atanackovic et al., 2024) learns amortized vector fields using graph neural networks but requires multiple datasets for training, limiting applicability when only a single time-series experiment is available.

**Geometric priors: spatial and structural alignment.** Spatial optimal transport methods leverage physical proximity to infer communication or alignment, using spatial transcriptomics measurements (Cang & Nie, 2020). For example, NicheFlow models microenvironment-mediated effects but assumes access to spatial coordinates (Sakalyan et al., 2025). This requirement is a practical barrier in the most common setting of dissociated scRNA-seq, where spatial coordinates are unavailable; in contrast, our approach enables communication-aware alignment *from dissociated scRNA-seq alone* by using curated ligand–receptor knowledge as a mechanistically grounded prior. More generally, Gromov–Wasserstein (GW) and Fused GW formulations compare samples via relational structure rather than raw features (Vayer et al., 2020). In single-cell applications, the relational structure is often instantiated as scalar similarity graphs capturing generic topology; such graphs are useful geometric summaries, but they do not encode *who signals to whom via which pathway*. Recent OT toolkits such as MosCOT provide scalable solvers for linear and fused OT, yet standard pipelines largely rely on feature-space or spatial distances, leaving the rich landscape of interaction-driven constraints unexplored (Klein et al., 2025). Related flow-based approaches incorporate geometric priors by restricting dynamics to the data manifold or a learned Riemannian metric (Huguet et al., 2022; Kapusniak et al., 2024).

**Communication priors (ours): directed interaction structure.** A large literature infers putative cell–cell communication from dissociated scRNA-seq using curated ligand–receptor catalogs and expression-based heuristics, providing biologically grounded *priors* on plausible signaling even without spatial coordinates. Tools such as CellPhoneDB systematically enumerate LR co-expression across cell types (Efremova et al., 2020). Meta-frameworks like LIANA+ unify and standardize CCI scoring across multiple LR resources and methods, facilitating method-agnostic comparisons and consensus analyses (Dimitrov et al., 2024). In our work, we derive a directed, typed interaction representation from ligand–receptor expression and inject it into an FGW objective, favoring couplings that encode smoothly evolving channel-specific signaling context across time. Relatedly, CytoBridge (Zhang et al., 2025) formulates an unbalanced mean-field Schrödinger bridge that learns interaction effects via a neural interaction potential alongside growth and transition dynamics, but it does not leverage *interpretable*, directed, typed ligand–receptor channels as an explicit coupling regularizer. Furthermore, because our contribution operates at the coupling level, it is agnostic to the choice of downstream continuous-time learner, unlike CytoBridge. In particular, `CellBRIDGE` can be used as a plug-and-play regularizer on top of the priors discussed above.

**Comparison with Related Works**   Table 4 provides a non-exhaustive comparison between `CellBRIDGE` and different methods across five key capabilities essential for modeling complex cellular dynamics. We define these criteria as follows:

- **Dynamic** indicates whether the method explicitly models temporal evolution across multiple experimental timepoints, as opposed to inferring dynamics or trajectories from a single static snapshot.

- **Trajectories** distinguishes methods that recover a continuous smooth path enabling predictions at unobserved intermediate timepoints from those that solely compute discrete couplings or transport maps between timepoints.

- **In-silico Perturbation** refers to the capability to perform principled interventions, allowing users to simulate and predict the system's response to specific stimuli or perturbations.

- **Structure-Aware** assesses whether the optimization objective explicitly models interactions between cells (e.g., via cell-cell communication or topological constraints) rather than treating cells as independent, isolated entities.

- **scRNA Data Sufficient** confirms whether the method can operate effectively using standard single-cell RNA sequencing inputs alone, without requiring auxiliary spatial transcriptomics data or multi-modal integration that are often unavailable.

*Table 4.* **Capability matrix for trajectory inference methods.** Comparison of graph-based, optimal transport, and continuous-time generative approaches for learning cellular dynamics from population snapshots. `CellBRIDGE` uniquely integrates typed interaction structure into transport-based trajectory inference using scRNA-seq data only.

| Method | Dynamic | Trajectories | In-silico Perturbation | Structure-Aware | scRNA Only | Mechanism | Example Reference |
|---|---|---|---|---|---|---|---|
| *Graph-Based and Pseudotime Methods* | | | | | | | |
| PAGA (Scanpy) | ✗ | ✗ | ✗ | ✗ | ✓ | Graph heuristics | Wolf et al. (2019) |
| Monocle / DPT | ✗ | ✗ | ✗ | ✗ | ✓ | Pseudotime graphs | Haghverdi et al. (2016) |
| scVelo | ✓ | ✗ | ✗ | ✗ | ✓ | RNA velocity | Bergen et al. (2020) |
| *Optimal Transport Alignment* | | | | | | | |
| Waddington-OT | ✓ | ~ | ✗ | ✗ | ✓ | Feature OT | Schiebinger et al. (2019) |
| SCOT | ✗ | ✗ | ✗ | ~ | ✗ | GW alignment | (Demetci et al., 2022) |
| MOSCOT | ✗ | ✗ | ✗ | ~ | ✓ | GW / FGW OT | (Klein et al., 2025) |
| *OT-Based Continuous-Time Dynamics* | | | | | | | |
| TrajectoryNet | ✓ | ✓ | ✗ | ✗ | ✓ | Neural ODE + OT | (Tong et al., 2020) |
| OT-CFM | ✓ | ✓ | ✗ | ✗ | ✓ | Flow matching | (Tong et al., 2024a) |
| Diffusion SB (DSB) | ✓ | ✓ | ✗ | ✗ | ✓ | Schrödinger bridge | (De Bortoli et al., 2021) |
| *Perturbation and Conditional Transport* | | | | | | | |
| CellOT | ✗ | ✗ | ✓ | ✗ | ✗ | Conditional OT | (Bunne et al., 2023) |
| NicheFlow / Spatial OT | ✗ | ✗ | ~ | ~ | ✗ | Spatial structure | (Sakalyan et al., 2025) |
| **`CellBRIDGE` (Ours)** | ✓ | ✓ | ✓ | ✓ | ✓ | Interaction-aware FGW + FM | – |

# B. Potential Applications of `CellBRIDGE`

Snapshots of cellular systems using single-cell RNA sequencing are now pervasive across diverse areas of biology and medicine. A few representative longitudinal datasets are summarized in Table 5. `CellBRIDGE` provides a principled framework to analyze such data by combining snapshot measurements with biologically typed ligand–receptor structure. This

*Table 5.* **Examples of public longitudinal single-cell datasets.** Each row summarizes a biomedical area, the available longitudinal single-cell setting, and representative references. The list is illustrative rather than exhaustive.

| Area | Dataset description | Representative references |
|---|---|---|
| Virology | Longitudinal PBMC or tissue scRNA-seq during viral infection, vaccination, or challenge studies, capturing early immune activation, peak response, and recovery. | Dengue virus: (Zanini et al., 2018); vaccine response: (Arunachalam et al., 2021) |
| Neurology | Brain single-cell time courses, including organoid and tissue systems that profile neuronal, glial, and immune-state changes over development or disease progression. | Brain organoids: (Camp et al., 2015) |
| Cardiology | Cardiac and vascular single-cell time courses following injury or during disease progression, capturing inflammation, remodeling, and repair. | Post-MI heart: (Farbehi et al., 2019); atherosclerosis: (Pan et al., 2020) |
| Immunology | Tissue and immune-cell scRNA-seq across baseline, inflammatory activation, and resolution or recovery in experimental model systems. | Lung inflammation: (Goldfarbmuren et al., 2020) |
| Development | Human iPSC or hPSC differentiation series that track lineage commitment, maturation, and cell-state transitions over multiple sampling stages. | Cardiomyocytes: (Strober et al., 2019); blood cells: (Tusi et al., 2018) |
| Regeneration | Injury-response time courses in organs such as liver, kidney, or muscle, capturing damage response, repair, and cellular remodeling. | Liver injury: (Chen et al., 2023) |

enables the reconstruction of coherent cell-state trajectories through optimal transport couplings and a learned continuous flow, as well as the exploration of counterfactual scenarios by selectively re-weighting interaction channels. The resulting outputs (shifts in lineage fate, changes in pathway usage, and differences in progression timing) offer interpretable readouts that can guide mechanistic hypotheses and help prioritize therapeutic strategies before experimental validation.

## C. Datasets

In addition to the synthetic dataset, we used 6 real-world scRNA datasets to showcase the effectiveness and limitations of our method. Details on the number of genes and the number of cells in each dataset can be found in Table 6.

*Table 6.* **Datasets used in our experiments.** Counts reflect the preprocessed objects used by `CellBRIDGE`. Time points indicate the observed sampling times for each dataset. The mouse cell atlas spans embryonic day E3.5 to E13.5.

| Dataset | Reference | Time points | #Cells | #Genes |
|---|---|---|---|---|
| Tumour | (Kortlever et al., 2017) | 0, 8, 24, 168 (h) | 31,536 | 22,681 |
| V1 Cortex | (Hrvatin et al., 2018) | 0, 1, 4 (h) | 6,505 | 17,008 |
| Dendritic Stimulus | (Shalek et al., 2014) | 0, 1, 2, 4, 6 (h) | 2,382 | 10,972 |
| Mouse embryo | (Moon et al., 2019) | 0, 6, 12, 18, 24 (d) | 18,203 | 17,789 |
| Macrophage Stimulus | (Wierenga et al., 2022) | 0, 3, 5 (h) | 223 | 478 |
| Mouse Cell Atlas | (Qiu et al., 2022) | E3.5 - E13.5 | 1.7 M | 29,452 |

### C.1. Synthetic example

In this section we detail the synthetic setup used in Section 5.1. We construct $\mathcal{D}_0$ as three 2D Gaussian clusters,

$$\mathcal{D}_0 = \bigcup_{k=0}^{2} \mathcal{S}_k, \qquad \mathcal{S}_k = \{X_i^{(k)}\}_{i=1}^{35}, \qquad X_i^{(k)} \overset{\text{i.i.d.}}{\sim} \mathcal{N}(\mu_k, 0.1\, I_2),$$

with centers $\mu_0 = (-2, 2)$, $\mu_1 = (0, 2)$, and $\mu_2 = (2, 2)$. The target snapshot $\mathcal{D}_1 = \bigcup_{k=0}^{2} \mathcal{S}_k'$ is obtained by translating each cluster via

$$T_0(x) = x + (4, -4), \quad T_1(x) = x + (0, -4), \quad T_2(x) = x + (-4, -4),$$

so that $\mathcal{S}_k' = \{T_k(X) : X \in \mathcal{S}_k\}$.

For structure, we define two-channel, directed relation tensors $G, G' \in \{0, 1\}^{105 \times 105 \times 2}$ over $\mathcal{D}_0$ and $\mathcal{D}_1$, respectively. Writing $G^{(c)}$ for channel $c$, we set

$$G^{(1)}_{ij} = \mathbf{1}\{X_i \in \mathcal{S}_1, \ X_j \in \mathcal{S}_0\}, \qquad G^{(2)}_{ij} = \mathbf{1}\{X_i \in \mathcal{S}_1, \ X_j \in \mathcal{S}_2\},$$

with $G'$ defined analogously on $\mathcal{D}_1$. Thus, channel 1 encodes $\mathcal{S}_1 \to \mathcal{S}_0$ and channel 2 encodes $\mathcal{S}_1 \to \mathcal{S}_2$.

### C.2. Lung Tumor

We use a private scRNA-seq dataset to study rapid tumour progression driven by RAS–MYC signalling using a *Kras*$^{\text{G12D}}$ lung tumour model with tamoxifen-inducible MycER. Samples were collected at 0 h (vehicle), 8 h, 24 h ($n = 8$ biological replicates per condition; 0 h is time zero). Lungs from LSL-*Kras*$^{\text{G12D}}$ (Jackson et al., 2001) and LSL-*Rosa26*$^{\text{MIE/MIE}}$ (*MycERT2*) mice (Murphy et al., 2008) were dissociated to single cells, red blood cells removed, filtered (70 $\mu$m), and 6,000 cells per sample were loaded for 10x Chromium $3'$ v3 libraries. Libraries were sequenced on a NovaSeq 6000 and processed with Cell Ranger v6.1.1 against `mm10`. All animal work complied with institutional ethical regulations of the Francis Crick Institute.

### C.3. V1 Cortex - Light stimulation

Adult (6–8 week) mice were dark-adapted for 7 days, then either euthanized in darkness (0h, control) or exposed to ambient light for 1h or 4h (Hrvatin et al., 2018). The visual cortex was profiled by scRNA-seq to capture early transcriptional responses to sensory input. We treat 0h as the source snapshot, 4h as the target snapshot, and use 1h as an intermediate time point for interpolation/validation.

### C.4. Dendritic-cell stimulus

We use the Shalek et al. (Shalek et al., 2014) dendritic-cell stimulus-response dataset, which profiles primary mouse bone-marrow-derived dendritic cells under innate immune stimulation. The dataset includes wild-type cells stimulated with LPS across early response time points, as well as matched knockout conditions used to dissect paracrine signalling. In our interpolation experiments, we use wild-type unstimulated cells as the source snapshot, wild-type LPS-stimulated cells at 4 h as the target snapshot, and intermediate LPS-stimulated time points for validation where applicable. For the perturbation analysis in Section F.11, we additionally use the experimentally observed 4 h knockout populations for *Ifnar1*, *Stat1*, and *Tnfr* as held-out perturbed targets.

### C.5. Macrophage stimulus

We use the single-cell RNA-seq dataset of Wierenga et al. (Wierenga et al., 2022), which profiles murine fetal liver-derived macrophages exposed to LPS with or without 24 h pre-treatment with docosahexaenoic acid (DHA, 25 $\mu$M). Cells were treated with LPS (20 ng/mL) and collected at 0 h, 1 h, and 4 h, then sequenced using the 10x Chromium platform. In our interpolation experiments, we use 0 h as the source snapshot, 4 h as the target snapshot, and 1 h as the held-out intermediate snapshot. Where condition-specific analyses are performed, cells are stratified by vehicle versus DHA pre-treatment before subsampling.

### C.6. Embryo development

In Section 5.4, we analyze a mouse embryoid body (EB) differentiation time course used in Moon et al. (2019), which profiles embryonic stem cells differentiating toward germ layers over 27 days by scRNA-seq. We use the first (Day 0) and third (Day 12) snapshot to infer the cellular dynamics, reserving data at Day 6 for interpolation/validation.

### C.7. Embryo cell atlas

To evaluate scalability on atlas-scale data and assess performance under challenging developmental dynamics, we additionally considered the mouse embryo cell atlas of (Qiu et al., 2022). We constructed two held-out interpolation tasks: E7.5→E8, holding out E7.75, and E7.75→E8.25, holding out E8. These transitions span rapid embryonic cell-state diversification and tissue remodeling, providing a challenging benchmark for trajectory inference methods.

# D. Experimental Details

In what follows, we provide details about our experiments presented in Section 5.

## D.1. Data pre-processing

Raw scRNA-seq files for all datasets were converted to AnnData to standardize processing. We applied basic QC, removing cells with $< 300$ detected genes and genes expressed in $< 3$ cells. Counts were library-size normalized per cell (fixed total), then log-normalized. We then selected the 2000 highly variable genes and computed a 20-component PCA on these features. Finally, we performed Harmony batch correction in PCA space (retaining both corrected and uncorrected embeddings for downstream analyses).

## D.2. Constructing CCIs using metacells

We detail how we construct CCIs using metacells in the ablation presented in Section 5.3. Without loss of generality and to keep the presentation simple (with matrix multiplications), we assume $K = 1$ (i.e., one LR pair) reducing the CCI tensors to matrices. Before constructing the CCI matrices, we cluster the cells in each snapshot using Leiden community detection on a $k$-nearest-neighbour (kNN) graph built from the PCA representations with Euclidean distances and $k = 10$. An example of the Leiden clustering with subsequent cell annotations is provided in Figure 5. We select the resolution $\rho^\star$ by scanning a small grid of resolutions and choosing the value whose *median* cluster size is closest to a target of $n^\star = 40$ cells.

Let $S \in \mathbb{R}_{\geq 0}^{n \times g}$ be the membership matrix of the resulting $g$ clusters (rows sum to 1 and correspond to one-hot assignments). We obtain metacell-level activations by averaging the $s_{cg}$ within clusters and form the metacell CCI in $\mathbb{R}^{g \times g}$ similarly as in the setting with individual cells.

Having constructed the metacell CCI matrix $\bar{G}$, we lift it back to the cell level via

$$\tilde{G} \;=\; S\,(S^\top S)^{-1}\,\bar{G}\,(S^\top S)^{-1}\,S^\top,$$

This lifting operation ensures $S^\top \tilde{G} S = \bar{G}$. In contrast to $G$, the matrix $\tilde{G}$ is constrained to lie in the subspace $\{SMS^\top \mid M \in \mathbb{R}^{g \times g}\}$, i.e., cell–cell interactions in $\tilde{G}$ are entirely mediated by metacell–metacell interactions.

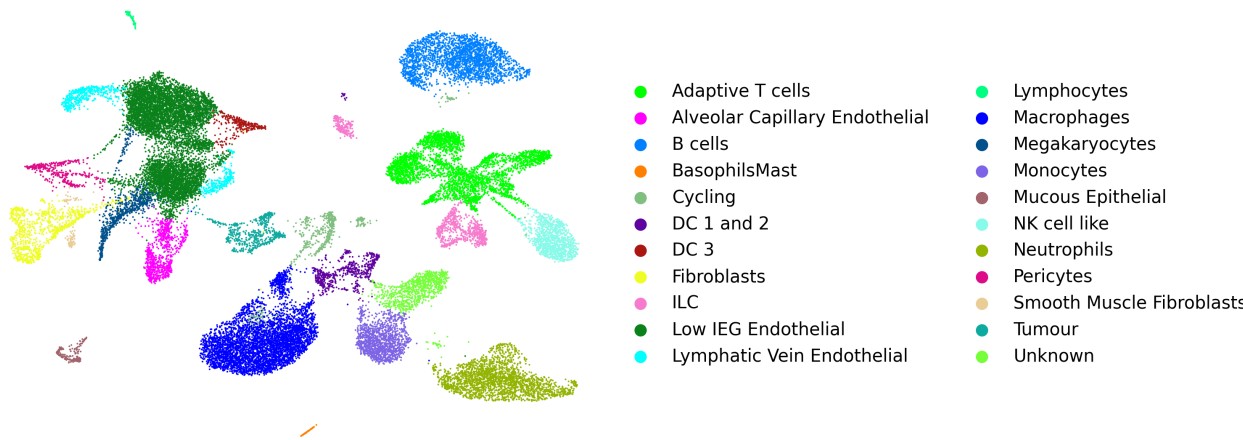

*Figure 5.* **Metacell construction example.** UMAP visualization of the single-cell RNA-seq data of the lung cancer dataset after Leiden clustering. Each point corresponds to an individual cell, colored by its assigned cluster and annotated with the corresponding cell type based on marker genes.

## D.3. Optimal transport solver

We extend POT's (Flamary et al., 2024) conditional-gradient (Frank–Wolfe) solver to handle multi-channel interactions. Given structure tensors $C_1 \in \mathbb{R}^{n_s \times n_s \times d}$, $C_2 \in \mathbb{R}^{n_t \times n_t \times d}$, marginals $p, q$ (uniform by default), and a matrix $\Sigma \succeq 0 \in \mathbb{R}^{d \times d}$, we measure discrepancies with the Mahalanobis norm $\|x\|_\Sigma = \sqrt{x^\top \Sigma^{-1} x}$.

Let $\langle A, B \rangle = \sum_{i,j} A_{ij} B_{ij}$ and write $C^{(r)}$ for the $r$-th channel of a structure tensor $C$. The GW quadratic term is

$$\mathcal{Q}(\Gamma) = \sum_{i,k,j,l} \left\| C_1[i,k] - C_2[j,l] \right\|_\Sigma^2 \Gamma_{ij} \Gamma_{kl} = \langle \mathrm{const}C, \Gamma \rangle - \langle \mathcal{B}(\Gamma), \Gamma \rangle,$$

with

$$\mathrm{const}C_{ij} = \sum_k \|C_1[i,k]\|_\Sigma^2 p_k + \sum_l \|C_2[j,l]\|_\Sigma^2 q_l, \qquad \mathcal{B}(\Gamma) = \sum_{r=1}^d C_1^{(r)} \, \Gamma \left( C_2^{(r)} \right)^\top.$$

The gradient computed by the solver is

$$\nabla \mathcal{Q}(\Gamma) = 2 \big( \mathrm{const}C - \mathcal{B}(\Gamma) \big) \tag{8}$$

We keep POT's CG loop, stopping criteria, and line-search options unchanged.

We minimize

$$\min_{\Gamma \in \Pi(p,q)} (1 - \alpha) \langle M, \Gamma \rangle + \alpha \mathcal{Q}(\Gamma),$$

with the same CG loop, where this objective is linearized using the gradient in Equation (8).

When $d = 1$ (scalar edges), the method reduces to the original POT solver.

### D.4. Normalization

To balance the contributions of the feature term and the structure term in the objective described at Equation (5), we rescale the feature cost matrix $C$ and the CCI tensors $G^{(0)}$ and $G^{(1)}$. We first compute the two endpoint couplings by solving the feature-only ($\alpha = 0$) and structure-only ($\alpha = 1$) problems, yielding $\Gamma_{\alpha=0}^\star$ and $\Gamma_{\alpha=1}^\star$. We then define the scaling factors as follows:

$$\Delta \mathcal{F} := \mathcal{F}(\Gamma_{\alpha=0}^\star) - \mathcal{F}(\Gamma_{\alpha=1}^\star), \tag{9}$$

$$\Delta \mathcal{S} := \mathcal{S}(\Gamma_{\alpha=1}^\star) - \mathcal{S}(\Gamma_{\alpha=0}^\star), \tag{10}$$

$$\tag{11}$$

and rescale the feature cost matrix and the CCI tensors:

$$C \leftarrow \frac{C}{|\Delta \mathcal{F}|}, \tag{12}$$

$$G^{(0)} \leftarrow \frac{G^{(0)}}{\sqrt{\Delta \mathcal{S}}}, \qquad G^{(1)} \leftarrow \frac{G^{(1)}}{\sqrt{\Delta \mathcal{S}}}. \tag{13}$$

This places the terms on comparable scales so that $\alpha$ meaningfully reflects the feature/structure trade-off, and increasing $\alpha$ from 0 to 1 smoothly interpolates between the Kantorovich and the Gromov–Wasserstein problems.

### D.5. Selection of ligand / receptor pairs

We apply LIANA's (Dimitrov et al., 2024) consensus rank aggregation with `expr_prop` = 0.1 to obtain per–cell-type interaction scores. We retain interactions with `cellphone_pvals` $\leq 0.05$ and `lr_logfc` $\geq 0$, then keep ligand–receptor pairs whose `expr_prod` exceeds the median within that significant set. We require the same significance criteria in each snapshot. For every surviving pair, we aggregate LIANA results across significant edges to compute the mean expression product, average specificity ranks, counts of significant source→target edges, and the numbers of unique source and target cell types. We define coverage as `coverage` = `n_edges`$/N_{\text{sig edges}}$ and retain only pairs with $0.10 \leq$ `coverage` $\leq 0.40$ and at least two sources and two targets. We compute a standardized score $s = 0.6 \, z(\text{mean\_expr}) + 0.4 \, z(-\text{spec\_rank})$ and greedily select pairs in descending $s$ while preventing repeated ligands or receptors. We keep the top 10 pairs for each dataset.

*Table 7.* **Flow-based baseline hyperparameters.** Baselines based on flow matching, diffusion bridges, or unbalanced optimal transport.

| Method | Category | Hyperparameter | Setting / Notes |
|---|---|---|---|
| **Diffusion Schrödinger Bridge (DSB)** | | | |
| DSB | Model | Score network | Encoder [16, 32], Decoder [64, 64, 64], latent dim 16 |
| | Training | IPF iterations | 10 outer IPF rounds |
| | | Optimisation steps | 10,000 gradient updates |
| | | Langevin steps | 12 per bridge trajectory |
| | | Batch size | 128 |
| | | Learning rate | $1 \times 10^{-4}$ |
| | Regularization | $\gamma$ schedule | $\gamma_{\min} = \gamma_{\max} = 10^{-3}$ |
| | | Mean matching | Enabled |
| | | EMA | Disabled |
| **MFM (Flow Matching + Riemannian Correction)** | | | |
| MFM | Velocity net | Architecture | MLP (hidden dim 64, depth 3) |
| | | Time embedding | Sinusoidal (dim 16) |
| | Training | Epochs | 500 |
| | | Batch size | 128 (train), 2048 (val) |
| | | Optimizer | AdamW (lr $= 10^{-3}$, wd $10^{-4}$) |
| | | Grad clipping | 1.0 |
| | GeoPath | Architecture | MLP (hidden dim 128, depth 3) |
| | | Activation | SELU |
| | | Optimizer | Adam (lr $= 10^{-4}$) |
| | Metric | LAND | $\gamma = 0.2$, $\rho = 10^{-3}$, $\alpha = 1.0$ |
| | | Max samples | 4096 |
| **SF2M** | | | |
| SF2M | Model | Velocity + score MLP | Hidden dim 64, depth 3 |
| | | Time embedding | Sinusoidal (dim 16) |
| | Distribution | $\sigma_{\text{bridge}}$ | 1.0 |
| | | $\sigma_{\text{sample}}$ | 1.0 |
| | Training | Epochs | 500 |
| | | Batch size | 128 (train), 2048 (val) |
| | | Optimizer | AdamW (lr $= 10^{-3}$, wd $10^{-4}$) |
| | | Grad clipping | 1.0 |
| **UOT-FM** | | | |
| UOT-FM | Model | Velocity MLP | Hidden dim 64, depth 3 |
| | | Time embedding | Sinusoidal (dim 16) |
| | Training | Epochs | 500 |
| | | Batch size | 128 (train), 2048 (val) |
| | | Optimizer | AdamW (lr $= 10^{-3}$, wd $10^{-4}$) |
| | OT | Convergence tol. | $10^{-9}$ (rel./abs.) |
| | | Marginal reg. | 1.0 |
| **Flow Matching** | | | |
| Flow Matching | Model | Velocity MLP | Hidden dim 64, depth 3 |
| | | Time embedding | Sinusoidal (dim 16) |
| | Training | Epochs | 500 |
| | | Batch size | 128 (train), 2048 (val) |
| | | Optimizer | AdamW (lr $= 10^{-3}$, wd $10^{-4}$) |

## D.6. Baselines

**Conditional Flow Matching hyperparameters** We detail the hyperparameters used for downstream in Table 7, which we kept fixed across the datasets. Given a 0.9/0.1 train/val split, we keep the checkpoint that minimizes the validation loss over the run.

**TrajectoryNet.** We use the implementation from the authors (Tong et al., 2020) available at `https://github.com/KrishnaswamyLab/TrajectoryNet`. We summarize the hyperparameters used in Table 8.

**Diffusion Schrodinger Bridges.** We use the implementation from the authors (De Bortoli et al., 2021) available at `https://github.com/JTT94/diffusion_schrodinger_bridge`. We summarize the hyperparameters used in Table 7.

*Table 8.* **Non–flow-based baseline hyperparameters.** Baselines not relying on flow matching or diffusion bridges.

| Method | Category | Hyperparameter | Setting / Notes |
|---|---|---|---|
| **MioFlow** | | | |
| | Model | Network layers | [64, 64, 64] |
| | Training | Learning rate | $1 \times 10^{-4}$ |
| | | Total epochs | 20 |
| | | Local epochs | 5 |
| | | Post-local epochs | 5 |
| | | Batch size | 256 |
| | | Batches / epoch | 100 |
| **Moscot** | | | |
| | OT | Entropic reg. ($\epsilon$) | 0.001 |
| | | Source reg. ($\tau_a$) | 1.0 |
| | | Target reg. ($\tau_b$) | 1.0 |
| **VGFM** | | | |
| | Model | Hidden dimension | 64 |
| | | Hidden layers | 3 |
| | | Activation | Tanh |
| | Training | Pre-train epochs | 300 |
| | | Training epochs | 50 |
| | | Batch size | 256 |
| | | Learning rate (init) | $1 \times 10^{-3}$ |
| | | Learning rate (second) | $1 \times 10^{-4}$ |
| | Solver | Step size | 0.01 |
| **TrajectoryNet** | | | |
| | Training | Iterations | 1000 |
| | | Batch size | 1000 |
| | | Learning rate | $1 \times 10^{-3}$ |
| | | Weight decay | $1 \times 10^{-5}$ |
| | Model | Architecture | 1 block, concatsquash layers (64–64–64) |
| | Regularization | $s_{\mathrm{L2int}}$ | $1 \times 10^{-3}$ |
| | | $k_{\mathrm{top}}$ | $1 \times 10^{-2}$ |
| | | Training noise | 0.1 |
| | ODE solver | Solver | dopri5 |
| | | Time scale | 0.4 (5 integration points) |
| | | Tolerances | rtol = atol = $10^{-5}$ |

**MIOFlow** We use the implementation from the authors (Huguet et al., 2022) available at `https://github.com/KrishnaswamyLab/MIOFlow`. We summarize the hyperparameters used in Table 8.

**Moscot** We use the implementation from the authors (Klein et al., 2025) available at `https://github.com/theislab/moscot`. We summarize the hyperparameters used in Table 8.

**VGFM** We use the implementation from the authors (Wang et al., 2025) available at `https://github.com/DongyiWang-66/VGFM`. We summarize the hyperparameters used in Table 8.

**MFM** We use the implementation from the authors (Kapusniak et al., 2024) available at `https://github.com/kkapusniak/metric-flow-matching`. We summarize the hyperparameters used in Table 7.

**SF2M** We developed a custom implementation of the SF2M framework (Tong et al., 2024b) to enable the integration of the `CellBRIDGE` structural prior into the simulation-free training objective. We summarize the hyperparameters used in Table 7.

**UOT-FM** To incorporate the interaction-aware coupling in an unbalanced setting, we utilized a custom implementation of UOT-FM based on the original formulation (Eyring et al., 2024). We summarize the hyperparameters used in Table 7.

## D.7. Lung cancer data experiment

For the experiment described in Section 5.3, we annotated the lung cancer dataset using canonical lineage and state markers (Table 9); an overview of the full dataset is shown in Fig. 5. Because whole-lung profiling dilutes treatment effects (the tumour comprises only a small fraction of total cells), we constructed a focused *tumour-niche* subset to increase sensitivity and interpretability. Concretely, we retained all tumour cells and subsampled an equal number of T cells, B cells, fibroblasts, and endothelial cells from the same specimens to form a minimal viable tumour microenvironment. We then reused the analysis pipeline described earlier with matched timepoints at 0 h, 8 h, and 24 h. The only modification was to the ligand–receptor (LR) library: for pathway-specific probes, we toggled custom LR pairs to mimic the presence or absence of a given ligand (e.g., EGFR) and quantified the resulting changes in inferred communication and downstream dynamics. Marker definitions are provided in Table 9, and a dot-plot confirming marker specificity and minimal cross-lineage leakage is shown in Fig. 6.

*Table 9.* Curated panel of positive marker genes used for per-cell scoring and assignment in the lung cancer dataset.

| Cell Type | Positive Markers |
|---|---|
| Differentiated AT1 | RTKN2, AGER |
| AT1 | CLDN18 |
| Tumour (AT2) | SFTPD, LAMP3, SCGB3A2 |
| Mucous Epithelial | DNAH12, AZGP1 |
| Endothelial | SEMA3G |
| Low IEG Endothelial | CDH5 |
| Alveolar Capillary Endothelial | EDNRB, RPRML |
| Lymphatic Vein Endothelial | LYVE1, SELE, VWF |
| Fibroblasts | COL1A2, PDGFRA |
| Smooth Muscle Fibroblasts | ACTA2, LGR6 |
| Fibroblast Subset | DCN |
| Pericytes | CSPG4 |
| Megakaryocytes | PPBP, PF4 |
| Erythrocytes | ALAS2 |
| Lymphocytes | CCL21A |
| Cycling | TOP2A |
| Neutrophils | S100A9, RETNLG |
| Basophils & Mast cells | MCPT8, MS4A2 |
| Macrophages | MARCO |
| Monocytes | LY6I |
| DC 1 and 2 | CLEC9A, XCR1, C1QA, SIGLECH |
| DC 3 | FSCN1, IL12B |
| NK cell like | NCR1, EOMES, TBX21 |
| ILC | RORA, RORC, IL2RA |
| Adaptive T cells | FOXP3, CD4, CD8A |
| B cells | CD79A |

### D.7.1. TUMOUR PROGRESSION QUANTIFICATION USING HALLMARK GENE SETS

There is no single, universally accepted definition of tumour progression. Clinical assessments typically use lesion size, extent of metastasis, and histopathology. While we observe distinct cellular changes and invasion over our 24 h window, these measures are not applicable at single-cell resolution. Instead, we construct an approximate *tumour differentiation* score based on the Hallmarks of Cancer (Hanahan & Weinberg, 2011), using the MSigDB *Hallmark* gene sets (Liberzon et al., 2015).

For each hallmark, we compute a per-cell score as the *median* expression across its member genes (chosen over the mean for robustness to sparsity and outliers). The overall progression score is then the mean across the 20 retained hallmarks. The full hallmark definitions are available in MSigDB (Liberzon et al., 2015); the selected hallmarks, their gene counts, and five example genes each are listed in Table 10. Hallmarks not applicable to our tumour context (e.g., hormonal signalling for breast/prostate, long-term metabolic programs) were excluded.

As a baseline check, we verify that tumour cells exhibit coherent changes along the selected hallmarks over 0 h→ 24 h; see Fig. 7.

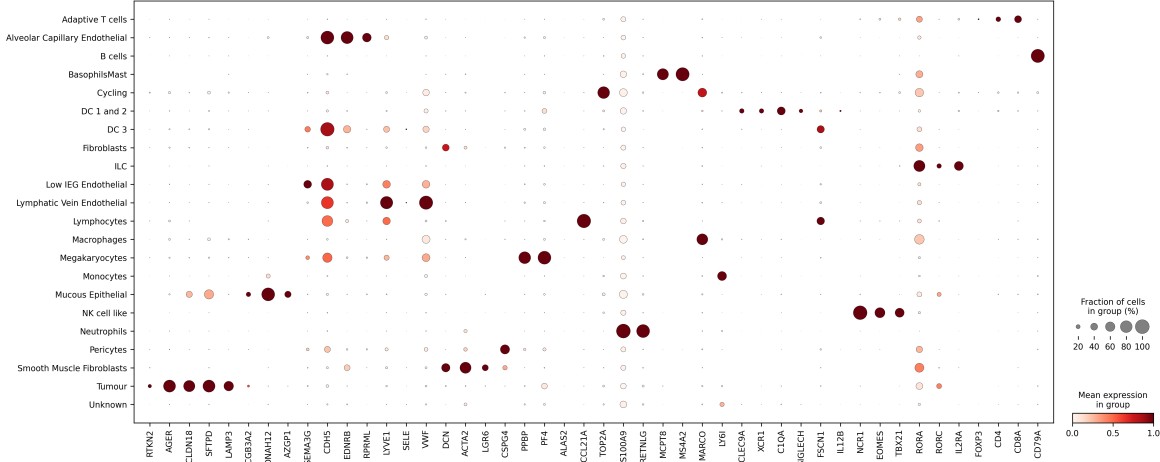

*Figure 6.* Dot-plot validation of curated marker genes across annotated cell types. Each column corresponds to a marker gene and each row to a cell-type label. Dot size encodes the fraction of cells expressing that gene, while color intensity represents its standardized expression level.

*Table 10.* **Hallmark gene sets used for trajectory summarization.** We list each set's size and five randomly sampled member genes.

| Gene set | # Genes | Random gene examples (5) |
|---|---|---|
| Angiogenesis | 36 | TIMP1, POSTN, VTN, THBD, NRP1 |
| Apoptosis | 161 | ERBB2, IL1B, DPYD, NEDD9, MADD |
| DNA Repair | 150 | GTF2B, RAE1, ADCY6, POLA2, TAF1C |
| E2F Targets | 200 | MCM7, PCNA, MCM4, RFC2, GINS1 |
| Epithelial–Mesenchymal Transition | 200 | SPP1, GPX7, LOX, THBS1, SLC6A8 |
| G2M Checkpoint | 200 | RBM14, AMD1, CDC27, UCK2, NDC80 |
| Glycolysis | 200 | SPAG4, PKP2, SLC25A13, PRPS1, ZNF292 |
| Hypoxia | 200 | S100A4, CSRP2, DTNA, PIM1, TPST2 |
| KRAS Signaling v1 | 200 | FSHB, YPEL1, BARD1, SLC6A3, ATP6V1B1 |
| KRAS Signaling v2 | 200 | CIDEA, KIF5C, LAT2, PDCD1LG2, PIGR |
| MYC Targets v1 | 200 | RAD23B, USP1, NAP1L1, NDUFAB1, SNRPA1 |
| MYC Targets v2 | 58 | PRMT3, AIMP2, SRM, EXOSC5, SUPV3L1 |
| Myogenesis | 200 | EIF4A2, PDE4DIP, ANKRD2, EPHB3, ATP6AP1 |
| Notch Signaling | 32 | SKP1, MAML2, HES1, FBXW11, DTX1 |
| Oxidative Phosphorylation | 200 | NDUFS8, VDAC1, UQCRQ, NDUFB3, NDUFB2 |
| p53 Pathway | 200 | TNNI1, SLC35D1, BTG1, FDXR, JAG2 |
| Peroxisome | 104 | IDH2, FIS1, EPHX2, SLC23A2, SLC25A4 |
| Reactive Oxygen Species Pathway | 49 | PRNP, OXSR1, SOD1, PDLIM1, TXN |
| TNF$\alpha$ Signaling via NF$\kappa$B | 200 | DUSP2, CEBPB, OLR1, CCL20, IL1A |
| Xenobiotic Metabolism | 200 | SSR3, HACL1, ARPP19, AHCY, GSR |

## D.8. Computational and memory costs

**Complexity of the full OT solver.** We solve Equation (5) with a custom conditional-gradient (Frank–Wolfe) solver detailed in Section D.3. Let $n_0$ and $n_1$ be the numbers of cells in the two snapshots and $K$ the number of ligand–receptor (LR) pairs (interaction channels).

Each Frank–Wolfe iteration consists of two main steps:

1. **Gradient computation.** This yields a per-iteration cost

$$\mathcal{O}\big(K\,n_0 n_1 (n_0 + n_1)\big)$$

because it requires performing the matrix multiplication of $C_1^{(r)}\Gamma$ and $\big(C_1^{(r)}\Gamma\big)\big(C_2^{(r)}\big)^\top$ for each channel $r$.

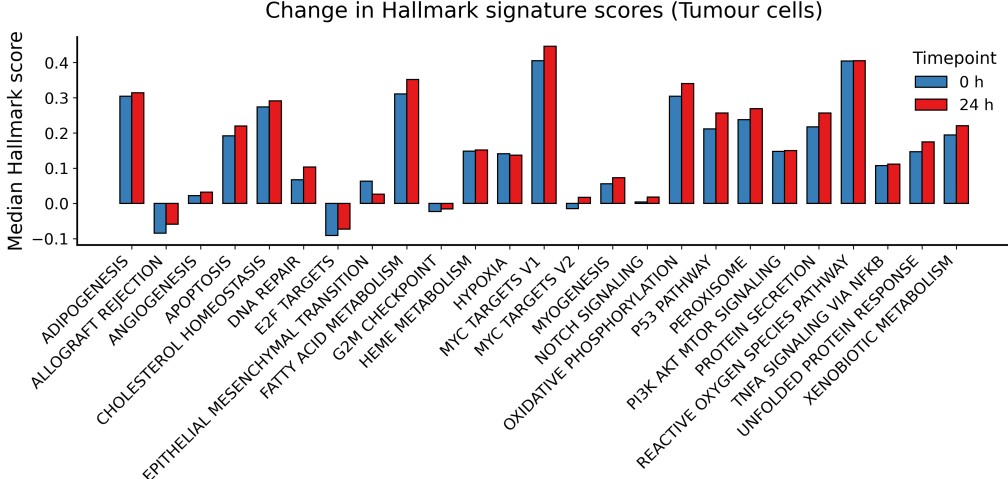

*Figure 7.* **Hallmark changes.** Changes in our dataset over 24 h following combined KRAS and MYC signalling across the 20 selected Hallmark gene sets.

*Table 11.* Wall-clock runtime (in seconds) of CellBRIDGE, decomposing into the OT part and the Flow matching part.

|         | Lung Tumour | V1 Light | Dendritic Stimulus | Mouse Cell Atlas |
|---------|-------------|----------|--------------------|------------------|
| OT [s]  | 211.3       | 107.0    | 4.3                | 2656             |
| FM [s]  | 189.1       | 186.0    | 24.2               | 4739             |

2. **Linear OT subproblem.** Given the linearized objective, we solve a linear OT problem over $\Pi(a, b)$ using POT's (Flamary et al., 2024) existing OT routine. Its complexity is

$$\mathcal{O}\big(\mathrm{cost}_{\mathrm{OT}}(n_0, n_1)\big),$$

(e.g. cubic in $n$ for a network-simplex LP, or $\mathcal{O}(T_{\mathrm{Sinkhorn}}\, n_0 n_1)$ for entropic OT).

If $T_{\mathrm{CG}}$ denotes the number of Frank–Wolfe iterations required to reach the desired tolerance, the total complexity of the CellBRIDGE OT stage is

$$\mathcal{O}\Big(T_{\mathrm{CG}}\big[K\, n_0 n_1 (n_0 + n_1) + \mathrm{cost}_{\mathrm{OT}}(n_0, n_1)\big]\Big).$$

**Wall-clock runtimes.** We report the wall-clock runtimes (seconds) in Table 11, decomposing it into the OT part (finding the coupling $\Gamma^*$) and the Flow matching part (fitting the velocity model).

**Memory footprint of the OT stage.** The dominant memory costs come from: (i) the coupling $\Gamma \in \mathbb{R}^{n_0 \times n_1}$, (ii) the feature cost matrix $C \in \mathbb{R}^{n_0 \times n_1}$, (iii) the multi-channel structure tensors $C_1 \in \mathbb{R}^{n_0 \times n_0 \times K}$ and $C_2 \in \mathbb{R}^{n_1 \times n_1 \times K}$ (corresponding to the CCI tensors $G^{(0)}$ and $G^{(1)}$), and (iv) a small number of auxiliary matrices of size $n_0 \times n_1$ (e.g. $\mathrm{const}C$, $B(\Gamma)$, and the gradient). Crucially, we never construct the full tensor of pairwise structure discrepancies in Equation (5). Instead, the structure term is implemented through the matrix products $C_1^{(r)} \Gamma \big(C_2^{(r)}\big)^\top$. As a result, the memory complexity of the OT solver scales as

$$\mathcal{O}\big(K(n_0^2 + n_1^2) + n_0 n_1\big),$$

Hence it is quadratic in the number of cells per snapshot and linear in the number of LR pairs $K$. In comparison, a feature-only OT solver ($\alpha = 0$) needs $C$ and $\Gamma$, with memory $\mathcal{O}(n_0 n_1)$.

**Using CellBRIDGE with large-scale datasets.** While the computational and memory cost remained reasonable across the datasets we used, for very large datasets it can be mitigated using standard scalable techniques that are orthogonal to our formulation:

- adding entropic regularization (Peyré & Cuturi, 2019) and using Sinkhorn-type solvers, which make the problem easier to optimize and reduce memory at the price of a small, controllable bias

- employing mini-batch optimization (Fatras et al., 2021), where the CCI prior is estimated from couplings computed on minibatches instead of the whole dataset

- constructing metacells, with details provided in Section D.2. Using metacells reduces the effective sample size. We evaluate this variant in Section 5.3.

These strategies preserve the form of the `CellBRIDGE` prior while substantially improving scalability for large-scale datasets.

## E. Downstream Dynamics with `CellBRIDGE`

**Overview.** This appendix details how the interaction-aware coupling produced by `CellBRIDGE` can be combined with different continuous-time trajectory models. Across all variants, the interaction prior is encoded exclusively in the cross-snapshot coupling $\Gamma^\star$ (or its induced joint distribution $\Pi$), while the downstream dynamics model determines how trajectories are parameterized between coupled endpoints. We consider four settings: deterministic flows via Conditional Flow Matching (CFM), geometry-aware flows via Metric Flow Matching (MFM), stochastic Schrödinger bridges via SF2M, and population-changing dynamics via unbalanced OT. Importantly, no modification of the interaction-aware FGW objective is required when switching between these mechanisms.

### E.1. Conditional Flow Matching (CFM)

We begin with Conditional Flow Matching (CFM), which serves as the simplest and default downstream instantiation of `CellBRIDGE`. CFM learns a deterministic, time-dependent velocity field whose induced flow matches a prescribed probability path between two endpoint distributions.

**Coupling-induced probability path.** Let $\rho_0$ and $\rho_1$ denote the empirical distributions associated with $\mathcal{D}_0$ and $\mathcal{D}_1$. Let $\Gamma^\star \in \mathbb{R}_+^{n_0 \times n_1}$ be the optimal coupling obtained from the interaction-aware FGW problem Equation (5), and define $M = \sum_{i,j} \Gamma^\star_{ij}$. We introduce the joint distribution

$$\Pi \;=\; \sum_{i=1}^{n_0} \sum_{j=1}^{n_1} \frac{\Gamma^\star_{ij}}{M} \, \delta_{(x_i, y_j)},$$

whose marginals are $\rho_0$ and $\rho_1$. For $t \in [0,1]$, we define the affine interpolation

$$Z_t = (1-t)X + tY, \qquad (X, Y) \sim \Pi,$$

and let $\rho_t = \mathcal{L}(Z_t)$. By construction, $\{\rho_t\}_{t \in [0,1]}$ forms a probability path connecting $\rho_0$ and $\rho_1$.

**Learning the velocity field.** We learn a time-dependent velocity field $v_\theta : \mathbb{R}^d \times [0,1] \to \mathbb{R}^d$ that generates the path $\{\rho_t\}$. For $(X, Y) \sim \Pi$ and $Z_t = (1-t)X + tY$, the interpolation implies a constant conditional drift

$$u_t(Z_t \mid X, Y) = Y - X.$$

We train $v_\theta$ by minimizing the Conditional Flow Matching objective

$$\mathcal{L}_{\text{CFM}}(\theta) = \mathop{\mathbb{E}}_{\substack{(X,Y) \sim \Pi \\ t \sim \text{Unif}[0,1]}} \left[ \left\| v_\theta(Z_t, t) - u_t(Z_t \mid X, Y) \right\|_2^2 \right] \tag{14}$$

$$= \mathop{\mathbb{E}}_{\substack{(X,Y) \sim \Pi \\ t \sim \text{Unif}[0,1]}} \left[ \left\| v_\theta(Z_t, t) - (Y - X) \right\|_2^2 \right]. \tag{15}$$

As shown in Lipman et al. (2024), the minimizer of this objective generates the target probability path. After training, trajectories are obtained by integrating the ODE $\dot{z}(t) = v_\theta(z(t), t)$ with initial condition $z(0) = x$.

### E.2. Unbalanced interaction-aware dynamics

We next describe how `CellBRIDGE` can be extended to settings where the total population mass changes between snapshots, e.g. due to cell proliferation or apoptosis. Rather than enforcing exact marginal constraints, we adopt an unbalanced OT formulation that relaxes mass conservation.

**Step 1: inferring non-uniform marginals.** We first ignore interaction structure and solve an unbalanced feature-only OT problem

$$\Gamma^{\mathrm{u}} \in \underset{\Gamma \in \mathbb{R}_{\geq 0}^{n_0 \times n_1}}{\arg\min} \left\{ \langle \Gamma, C \rangle + \lambda_0 \,\mathrm{KL}(\Gamma \mathbf{1} \,\|\, a) + \lambda_1 \,\mathrm{KL}(\Gamma^{\top} \mathbf{1} \,\|\, b) \right\}, \tag{16}$$

where $\mathbf{1}$ denotes the all-ones vector. From the optimal solution we extract the reweighted marginals

$$\tilde{a} = \Gamma^{\mathrm{u}} \mathbf{1}, \qquad \tilde{b} = (\Gamma^{\mathrm{u}})^{\top} \mathbf{1},$$

which are renormalized to sum to one. This step is independent of the interaction weight $\alpha$, allowing the same marginals to be reused across different FGW trade-offs.

**Step 2: interaction-aware FGW with frozen marginals.** In a second step, we fix $(\tilde{a}, \tilde{b})$ and solve the interaction-aware FGW problem

$$\min_{\Gamma \in \Pi(\tilde{a}, \tilde{b})} (1 - \alpha)\,\mathcal{F}(\Gamma) + \alpha\,\mathcal{S}(\Gamma), \tag{17}$$

where $\mathcal{F}$ and $\mathcal{S}$ are defined as in Equation (5). The resulting coupling preserves multi-LR-pair interaction structure while allowing unequal total mass between snapshots. This coupling can be passed unchanged to any downstream dynamics model, including CFM, MFM, or SF2M.

### E.3. Combining `CellBRIDGE` with Metric Flow Matching (MFM)

We now describe how the interaction-aware coupling produced by `CellBRIDGE` can be combined with Metric Flow Matching (MFM) (Kapusniak et al., 2024). MFM generalizes CFM by encouraging trajectories to follow geodesics of a data-dependent Riemannian metric $g$.

Given a coupling $q$ between endpoint distributions, MFM first learns interpolants

$$x_{t,\eta} = (1 - t)x_0 + tx_1 + t(1 - t)\phi_{t,\eta}(x_0, x_1),$$

by minimizing the geodesic energy

$$L_g(\eta) = \mathbb{E}_{(x_0, x_1) \sim q,\, t} \left[ \dot{x}_{t,\eta}^{\top} G(x_{t,\eta}; \mathcal{D}) \dot{x}_{t,\eta} \right],$$

where $G(\cdot; \mathcal{D})$ denotes the coordinate representation of $g$. In our experiments we use the LAND metric $g_{\mathrm{LAND}}$ (Arvanitidis et al., 2016).

After fitting $\eta^{\star}$, we train a velocity field using the interaction-aware coupling $\Pi$ via

$$\mathcal{L}_{\mathrm{MFM}}(\theta, \eta) = \mathbb{E}_{\substack{(X, Y) \sim \Pi \\ t \sim \mathrm{Unif}[0,1]}} \left[ \left\| v_\theta(Z_{t,\eta}, t) - \dot{x}_{t,\eta}(X, Y) \right\|_{g(Z_{t,\eta})}^2 \right], \tag{18}$$

where $\| \cdot \|_{g(Z_{t,\eta})}$ denotes the norm induced by the metric at $Z_{t,\eta}$.

### E.4. Combining `CellBRIDGE` with SF2M

Finally, we describe how `CellBRIDGE` can be combined with SF2M (Tong et al., 2024b) to obtain interaction-aware Schrödinger-bridge dynamics. SF2M learns both a drift $v_\theta$ and a score $s_\theta$ by regressing to the conditional drift and score of a mixture of Brownian bridges between coupled endpoints.

For a single bridge with diffusion $\sigma$, the conditional marginal at time $t \in (0, 1)$ is

$$p_t(x \mid x_0, x_1) = \mathcal{N}\big(x;\, (1 - t)x_0 + tx_1,\, \sigma^2 t(1 - t)I_d\big),$$

with closed-form drift and score. To combine **SF2M** with `CellBRIDGE`, we simply replace the entropic OT endpoint coupling with the interaction-aware coupling $\Pi$. Training samples are generated by

$$t \sim \text{Unif}[0,1], \qquad (X,Y) \sim \Pi, \qquad Z_t \sim p_t(\cdot \mid X, Y).$$

The resulting SF2M objective is

$$\mathcal{L}_{\text{SF}^2\text{M}}(\theta) = \mathbb{E}\Big[\|v_\theta(t, Z_t) - u_t^\circ(Z_t \mid X, Y)\|_2^2 \tag{19}$$

$$+ \lambda(t)^2 \|s_\theta(t, Z_t) - \nabla_z \log p_t(Z_t \mid X, Y)\|_2^2\Big], \tag{20}$$

where $\lambda(t) = 2\sqrt{t(1-t)}/\sigma$.

## F. Additional Results

### F.1. Synthetic dataset

On synthetic data, where the ground-truth is known, we also evaluate direct matching metrics (Hits@1 and Transport Rank Error (TRE)). The results are reported in Figure 8.

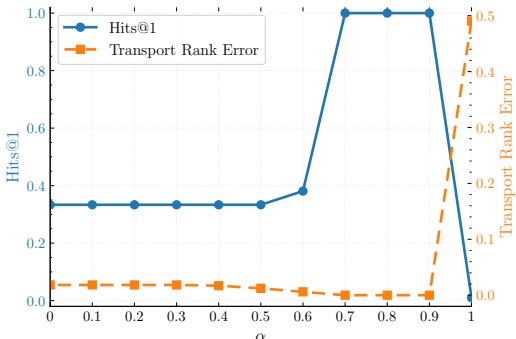

*Figure 8.* **Structure-aware coupling recovers the ground-truth transport map on the synthetic dataset.**

### F.2. Stimulus datasets

We reproduce the experimental setup described in Section 5.1 and Section 5.2 with the macrophage stimulus-response dataset. We report the results in Figure 9 and Table 12, which are consistent with the findings on the other datasets.

*Table 12.* **Interpolation error for continuous time dynamics (lower is better).** `CellBRIDGE` with varying structure weight $\alpha$ vs. baselines for the macrophage stimulus datasets. We report mean±std over 5 runs. MIOFlow failed to converge on stimulus CPG (numerical instability).

| Method | $\alpha$ | Stimulus PIC | | Stimulus CPG | | Stimulus LPS | | Stimulus PCSK3 | |
|---|---|---|---|---|---|---|---|---|---|
| | | $W_1$ | $W_2$ | $W_1$ | $W_2$ | $W_1$ | $W_2$ | $W_1$ | $W_2$ |
| TrajectoryNet | — | $5.628\,(0.055)$ | $5.930\,(0.049)$ | $5.361\,(0.085)$ | $5.826\,(0.080)$ | $5.087\,(0.109)$ | $5.589\,(0.078)$ | $5.033\,(0.051)$ | $5.434\,(0.051)$ |
| DSB | — | $5.796\,(0.574)$ | $5.833\,(0.571)$ | $4.500\,(0.128)$ | $4.594\,(0.114)$ | $4.685\,(0.533)$ | $4.815\,(0.528)$ | $4.648\,(0.328)$ | $4.749\,(0.324)$ |
| OT-CFM | — | $5.544\,(0.038)$ | $5.614\,(0.036)$ | $4.430\,(0.019)$ | $4.512\,(0.020)$ | $4.434\,(0.052)$ | $4.582\,(0.058)$ | $4.423\,(0.020)$ | $4.551\,(0.018)$ |
| OT-MFM | — | $5.501\,(0.020)$ | $5.566\,(0.022)$ | $4.464\,(0.013)$ | $4.556\,(0.015)$ | $4.440\,(0.033)$ | $4.592\,(0.038)$ | $4.384\,(0.008)$ | $4.505\,(0.008)$ |
| UOT-FM | — | $5.414\,(0.027)$ | $5.492\,(0.024)$ | $4.572\,(0.033)$ | $4.733\,(0.045)$ | $4.570\,(0.122)$ | $4.785\,(0.143)$ | $4.332\,(0.007)$ | $\mathbf{4.483\,(0.008)}$ |
| SF2M | — | $6.132\,(0.059)$ | $6.212\,(0.058)$ | $4.959\,(0.044)$ | $5.053\,(0.043)$ | $4.930\,(0.051)$ | $5.057\,(0.054)$ | $5.048\,(0.031)$ | $5.155\,(0.032)$ |
| VGFM | — | $9.796\,(0.658)$ | $9.863\,(0.683)$ | $8.242\,(0.192)$ | $8.362\,(0.205)$ | $8.026\,(0.135)$ | $8.158\,(0.1469)$ | $8.205\,(0.1101)$ | $8.297\,(0.110)$ |
| MIOFlow | — | $9.365\,(0.382)$ | $9.440\,(0.404)$ | $16899\,(22805)$ | $22007\,(31264)$ | $7.807\,(0.038)$ | $7.927\,(0.034)$ | $8.179\,(0.150)$ | $8.281\,(0.166)$ |
| Moscot | — | $7.213\,(0.000)$ | $7.244\,(0.0000)$ | $6.983\,(0.000)$ | $7.087\,(0.000)$ | $7.663\,(0.000)$ | $7.786\,(0.000)$ | $6.734\,(0.000)$ | $6.832\,(0.000)$ |
| CellBRIDGE+SF2M | 0.5 | $6.109\,(0.081)$ | $6.193\,(0.086)$ | $4.948\,(0.066)$ | $5.027\,(0.063)$ | $4.846\,(0.046)$ | $4.973\,(0.046)$ | $4.960\,(0.058)$ | $5.084\,(0.058)$ |
| | 1 | $6.110\,(0.062)$ | $6.197\,(0.063)$ | $4.951\,(0.056)$ | $5.037\,(0.058)$ | $4.874\,(0.068)$ | $5.016\,(0.065)$ | $5.016\,(0.029)$ | $5.173\,(0.039)$ |
| CellBRIDGE+MFM | 0.5 | $5.483\,(0.012)$ | $5.544\,(0.014)$ | $4.448\,(0.007)$ | $4.527\,(0.006)$ | $4.442\,(0.035)$ | $4.589\,(0.041)$ | $4.386\,(0.016)$ | $4.520\,(0.013)$ |
| | 1 | $5.376\,(0.021)$ | $\mathbf{5.440\,(0.022)}$ | $4.460\,(0.079)$ | $4.543\,(0.082)$ | $4.477\,(0.050)$ | $4.641\,(0.062)$ | $4.329\,(0.023)$ | $4.507\,(0.029)$ |
| CellBRIDGE+UOT-FM | 0.5 | $5.355\,(0.021)$ | $5.451\,(0.018)$ | $4.544\,(0.033)$ | $4.700\,(0.035)$ | $4.489\,(0.054)$ | $4.692\,(0.061)$ | $4.343\,(0.025)$ | $4.522\,(0.033)$ |
| | 1 | $\mathbf{5.346\,(0.024)}$ | $5.487\,(0.038)$ | $4.547\,(0.035)$ | $4.755\,(0.040)$ | $4.489\,(0.038)$ | $4.716\,(0.041)$ | $\mathbf{4.325\,(0.038)}$ | $4.531\,(0.038)$ |
| CellBRIDGE+CFM | 0.5 | $5.490\,(0.018)$ | $5.555\,(0.019)$ | $\mathbf{4.427\,(0.021)}$ | $\mathbf{4.502\,(0.025)}$ | $\mathbf{4.380\,(0.021)}$ | $\mathbf{4.517\,(0.021)}$ | $4.396\,(0.023)$ | $4.531\,(0.019)$ |
| | 1 | $5.446\,(0.018)$ | $5.512\,(0.016)$ | $4.440\,(0.041)$ | $4.518\,(0.044)$ | $4.431\,(0.126)$ | $4.577\,(0.141)$ | $4.352\,(0.020)$ | $4.530\,(0.033)$ |

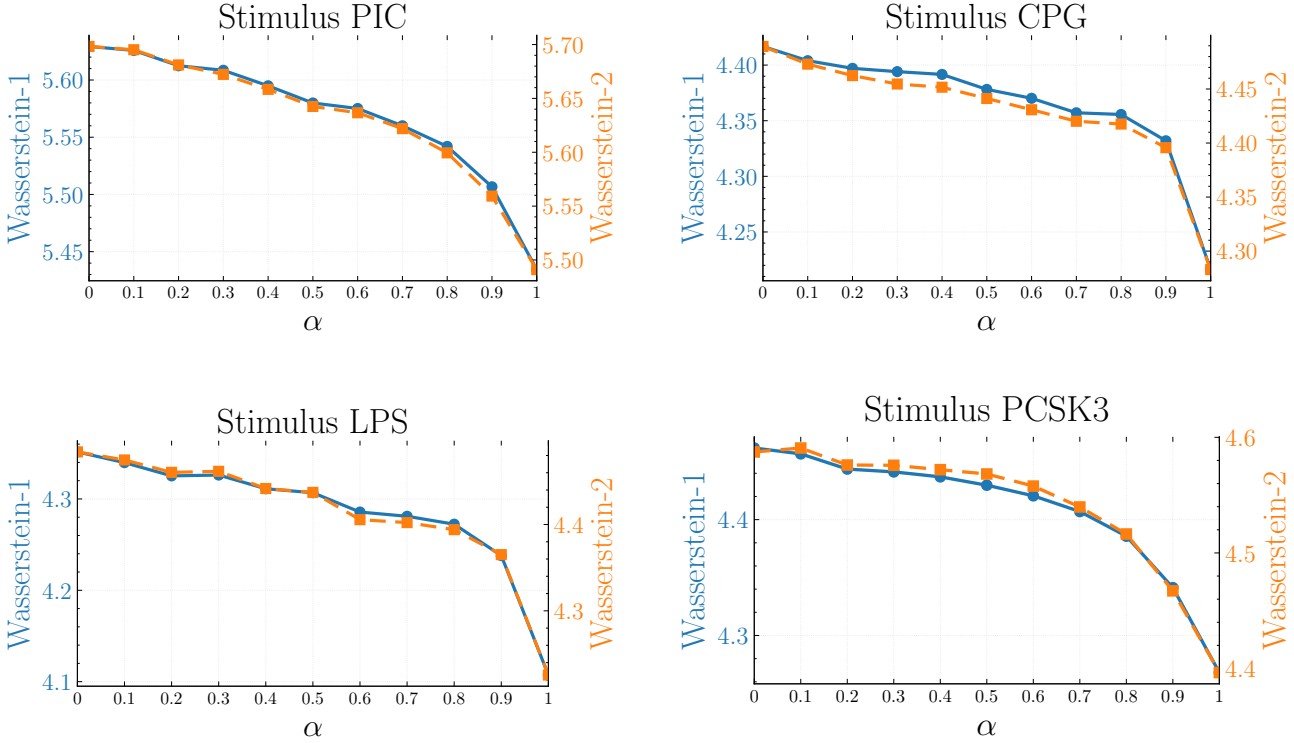

*Figure 9.* **Interpolation** results for the macrophage stimulus datasets.

### F.3. Scaling `CellBRIDGE` to cell atlas level datasets

We extended the evaluation to substantially larger datasets with two additional interpolation tasks on a mouse development cell atlas (Qiu et al., 2022). These results show that CellBRIDGE remains feasible in this larger-scale regime (see Table 13). We did not include UOT-FM on this benchmark because the unbalanced OT solver did not scale to datasets of this size in our experiments.

*Table 13.* **Interpolation error for embryo developmental transitions (lower is better).** We report mean $\pm$ std over runs for $W_1/W_2$.

| | E7.5→E8 | | E7.75→E8.25 | |
| Method | $W_1$ | $W_2$ | $W_1$ | $W_2$ |
|---|---|---|---|---|
| OT-CFM | $3.266_{(0.011)}$ | $3.575_{(0.023)}$ | $3.229_{(0.018)}$ | $\mathbf{3.249_{(0.010)}}$ |
| OT-MFM | $3.249_{(0.007)}$ | $3.586_{(0.025)}$ | $3.235_{(0.015)}$ | $3.250_{(0.007)}$ |
| SF2M | $4.967_{(0.078)}$ | $5.382_{(0.078)}$ | $4.748_{(0.108)}$ | $4.884_{(0.125)}$ |
| CellBRIDGE+CFM | $3.068_{(0.007)}$ | $3.555_{(0.005)}$ | $\mathbf{3.131_{(0.022)}}$ | $3.323_{(0.037)}$ |
| CellBRIDGE+MFM | $\mathbf{3.054_{(0.009)}}$ | $\mathbf{3.546_{(0.014)}}$ | $3.140_{(0.023)}$ | $3.384_{(0.020)}$ |
| CellBRIDGE+SF2M | $4.585_{(0.110)}$ | $4.872_{(0.132)}$ | $4.767_{(0.238)}$ | $4.947_{(0.225)}$ |

### F.4. Sensitivity of couplings to catalog edits

The experiment presented in Section 5.3 involved perturbing the LR catalog by removing specific LR pairs. In Table 14, we show how the coupling changes, by computing the fraction of source cells whose target argmax differs between "active" vs. "inactive" LR libraries for each pathway.

### F.5. Comparing cell interaction types

We further examined how different classes of molecular interactions influence the resulting transport couplings. Using our automated selection procedure (Section D.5), we identified a top-ranking set of 10 ligand-receptor pairs for each of

*Table 14.* **Coupling changes (argmax) at** $\alpha = 1.0$. Fraction of source cells whose target argmax differs between "active" vs. "inactive" LR libraries for each pathway; $N=2195$ source cells. Targeted pathways (EGFR/ALK/MET) show large shifts, while cardio–renal controls (RAAS, Vasopressin, Natriuretic) show little or moderate effect, as expected.

| Pathway / System | Coupling changed (count / $N$) | Percent |
|---|---|---|
| EGFR (targeted) | 2071/2195 | 94.35% |
| ALK (targeted) | 2164/2195 | 98.59% |
| MET (targeted) | 2154/2195 | 98.13% |
| RAAS (control) | 0/2195 | 0.00% |
| Vasopressin (control) | 0/2195 | 0.00% |
| Natriuretic (control) | 1582/2195 | 72.07% |

the datasets. We contrasted this against a matched set of 10 canonical long-range soluble cytokines and growth factors: (CXCL12-CXCR4, VEGFA-KDR, CCL5-CCR5, TGFB1-TGFBR2, IL6-IL6R, EGF-EGFR, TNF-TNFRSF1A, IGF1-IGF1R, CSF1-CSF1R, IFNG-IFNGR1). As shown in Table 15, the cytokine pairs exhibit slightly higher Wasserstein ($W_1$ and $W_2$) distances compared to the results obtained previously with our selection procedure. This suggests that the specific interaction modes we keep have more informative topological constraints on the transport map than generic diffusive signaling, effectively recovering structure-aware couplings that reflect the physical tissue architecture.

*Table 15.* **Wasserstein distances for pure structural alignment** ($\alpha = 1$). Comparison of interpolation performance using generic Long Range priors versus our automated selection procedure. Lower values indicate better alignment.

| Dataset | Interaction Prior | $\mathcal{W}_1 \downarrow$ | $\mathcal{W}_2 \downarrow$ |
|---|---|---|---|
| **V1 Light** | Long range | 2.42 | 2.63 |
| | Dataset-specific | **2.35** | **2.59** |
| **Immune** | Long range | **3.58** | **3.73** |
| | Dataset-specific | 3.59 | **3.73** |
| **Lung Cancer** | Long range | 2.10 | 2.33 |
| | Dataset-specific | **2.02** | **2.30** |

## F.6. Combining `CellBRIDGE` with other priors

A key advantage of `CellBRIDGE` is its modularity: the CCI-derived prior only depends on the CCI tensors $(G^{(0)}, G^{(1)})$ and on a coupling $\Gamma$, and is therefore largely orthogonal to how $\Gamma$ is obtained. As a consequence, the CCI prior can be combined with a wide range of existing priors or architectural choices for trajectory inference. Here, we illustrate this flexibility by extending `CellBRIDGE` to two settings: (i) unbalanced OT, which explicitly accounts for cell birth and death between snapshots, and (ii) metric flow matching, which replaces the standard Euclidean flow-matching objective with a geometry-aware variant. Details on both of these implementations can be found in Section E.2 and Section E.3.

We reproduce the experiment in Section 5.1 with these `CellBRIDGE` variants, and report the results in Figure 10 and Figure 11. We notice the following:

- $\alpha > 0$ **remains optimal.** For all datasets and the two `CellBRIDGE` variants, the best $W_1/W_2$ values occur at a non-zero structure weight $\alpha$, mirroring the behavior observed in Section 5.1.

- **Complementary to other priors.** The fact that $\alpha > 0$ remains optimal shows that adding the CCI prior on top of MFM or UOT-FM yields consistent improvements over the corresponding feature-only baselines, highlighting that `CellBRIDGE`'s gains are not tied to a specific OT or flow-matching objective, but rather come from the biological prior.

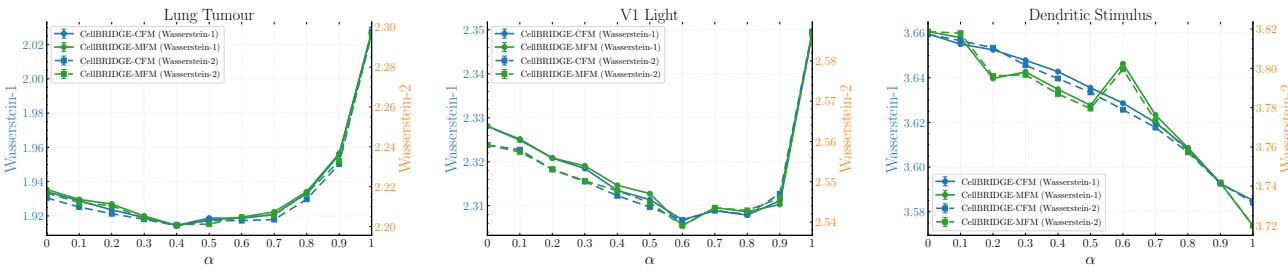

*Figure 10.* **Incorporating the CCI prior with MFM.** We plot the $W_1$ and $W_2$ distances between the interpolated and empirical $t_1$ snapshots as $\alpha$ varies.

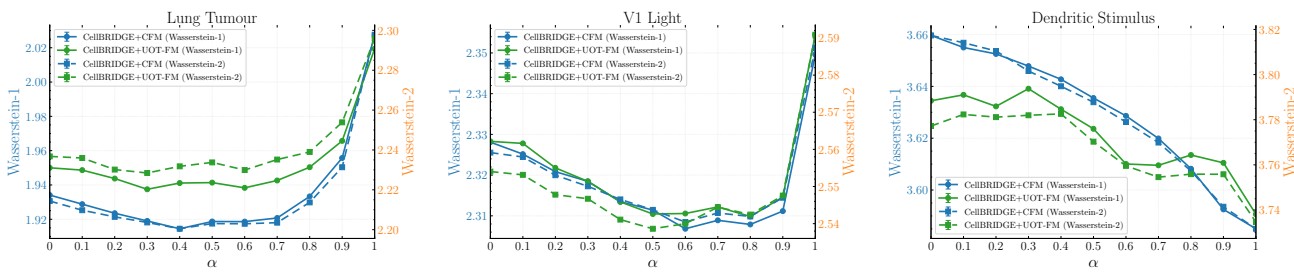

*Figure 11.* **Incorporating the CCI prior with UOT-CFM.** We plot the $W_1$ and $W_2$ distances between the interpolated and empirical $t_1$ snapshots as $\alpha$ varies.

### F.7. Sensitivity analysis on the LR expressions

In this section, we study the sensitivity of `CellBRIDGE` to measurement noise in the LR expressions. We inject this noise in LR genes expression by adding zero-mean Gaussian noise to the gene expressions before applying the Hill transform and clipping below by 0, i.e. we define $\tilde{x}_{cg} = \max(0, x_{cg} + \epsilon_{cg})$ where $\epsilon_{cg} \sim \mathcal{N}(0, \sigma_g^2)$. $\sigma_g^2$ denotes a gene-specific noise variance, defined as $\sigma_g = \beta\hat{\sigma}_g$, with $\hat{\sigma}_g$ the empirical standard deviation of $\{x_{cg} \mid c \in [n]\}$ (to take into account per-gene variance) and $\beta$ a scaling factor. From these perturbed expressions, we compute the activations $\tilde{s}_{cg}$ and we construct the CCI tensors with the entries $\tilde{q}_{i \to j}^{(p_k)} = \tilde{s}_{i\ell_k}\,\tilde{s}_{jr_k}$ and obtain the couplings by solving Equation (5).

We report the results in Figure 12, where we sweep $\beta$ for different values across the interval $[0, 2]$, with $\alpha = 1$.

As the noise scale $\beta$ increases, both $W_1$ and $W_2$ gradually deteriorate across all three datasets. This non-zero sensitivity is expected and desirable: if the CCI prior was irrelevant, corrupting the LR expressions would leave the interpolation error unchanged. Instead, adding noise worsens alignment, showing the benefits of the prior. Furthermore, the performance is relatively robust to small levels of noise for the Lung tumour and V1 Light datasets. Interestingly, for the V1 Light dataset, we see that $\beta \in \{0.1, 0.2\}$ improves the results upon $\beta = 0$, which we attribute to a small regularization / denoising effect. Adding a small amount of centered Gaussian noise before the Hill transform and clipping makes low-intensity ligand or receptor expressions become zero while leaving strongly expressed pairs essentially unchanged. The results are noisier for the Dendritic Stimulus dataset, which we attribute to the smaller size of the dataset.

### F.8. Sensitivity with respect to $K_g$ and $h_g$

In this section, we conduct a sensitivity analysis on the hyperparameters $K_g$ and $h_g$, used to define the interaction scores in Section 4.1 as $s_{cg} = x_{cg}^{h_g}/(x_{cg}^{h_g} + K_g^{h_g})$. We consider different values of the percentile level $p \in \{80, 90, 99\}$ (with $K_g = Q_g(p)$ denoting the $p$-th percentile of $\{x_{cg} \mid c \in [n]\}$) and $h_g \in \{1, 2, 4\}$, for $\alpha = 0.5$. We report the interpolation results in Figure 13. We observe that the performance is largely insensitive to the specific choice of these parameters. This stability justifies the use of standard default values (90th percentile and $h_g = 1$) across our experiments without the need for extensive per-dataset tuning.

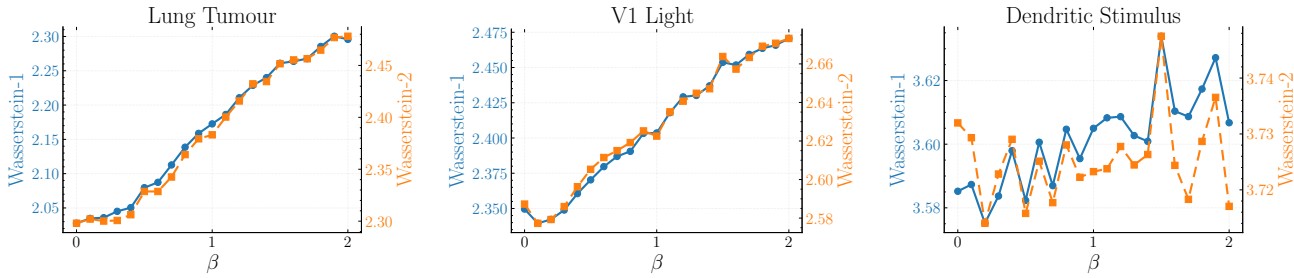

*Figure 12.* **Perturbation of the LR expressions with Gaussian noise.** We plot the $W_1$ and $W_2$ distances between the interpolated and empirical $t_1$ snapshots as the scaling factor of the noise $\beta$ increases.

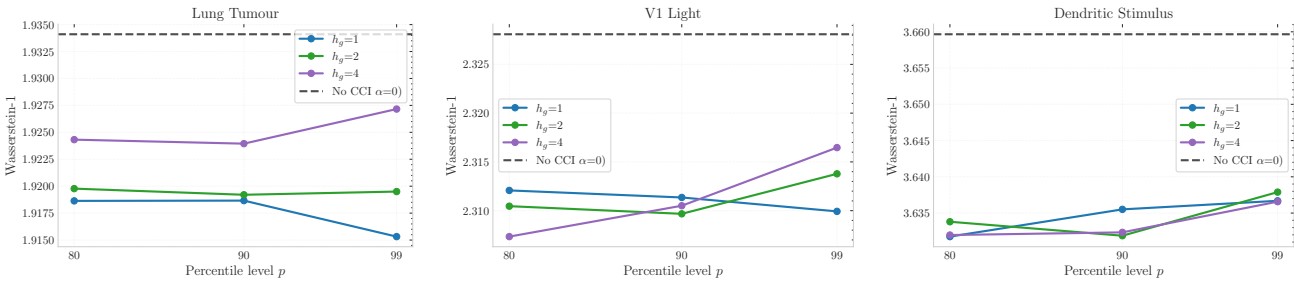

*Figure 13.* **Sensitivity analysis on the hyperparameters of the Hill transform.** We plot the $W_1$ distances between the interpolated and empirical $t_1$ snapshots for different values of $K_g$ and $h_g$.

## F.9. Path curvature analysis

To empirically demonstrate that `CellBRIDGE` learns non-linear interaction effects, we measure the average path length ratio (displacement divided by path length) of the inferred trajectories:

$$S(z_0, v_\theta) = \frac{\|z_1 - z_0\|_2}{\int_0^1 \|v_\theta(z_t, t)\|_2\, dt} \tag{21}$$

where $z_t = z_0 + \int_0^t v_\theta(z_t, t)dt$ for $t \in [0, 1]$ and $z_0$ denotes the initial point.

A ratio of 1.0 indicates a straight line, while values $< 1.0$ indicate curvature.

As shown in Table 16, increasing the interaction weight $\alpha$ leads to significantly higher curvature (lower ratios), confirming that incorporating interactions prevents the model from simply learning independent straight lines.

*Table 16.* Path length ratio comparison across different datasets.

| $\alpha$ | Lung Tumour | Dendritic Stimulus | V1 Light |
|---|---|---|---|
| 0 | $0.974 \pm 0.001$ | $0.982 \pm 0.002$ | $0.954 \pm 0.004$ |
| 0.5 | $0.952 \pm 0.002$ | $0.979 \pm 0.003$ | $0.907 \pm 0.011$ |
| 1 | $0.862 \pm 0.011$ | $0.969 \pm 0.005$ | $0.635 \pm 0.009$ |

## F.10. Normalization procedure ablation

Our normalization scheme described in Section D.4 requires solving two OT problems (for $\alpha = 0$ and $\alpha = 1$) to calibrate the relative scales of the cost and structural terms. The motivation is that $\alpha$ should smoothly control the balance between the two terms. We test a simpler normalization strategy on the tumour dataset that avoids these endpoint OT solves, by scaling $C$, $G^{(0)}$, and $G^{(1)}$ by their respective medians. The results in Figure 14 show that our normalization provides better calibration between the feature and structural terms.

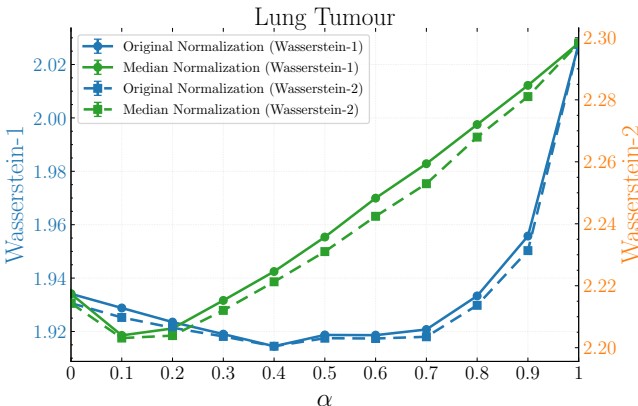

*Figure 14.* Comparison between our proposed normalization schemes and a median-based normalization for the Lung Tumour dataset, evaluated via held-out interpolation as in Section 5.1.

### F.11. Perturbation analysis on the dendritic-cell stimulus dataset

To test whether `CellBRIDGE` can model signalling-level perturbations beyond the lung tumour setting, we performed an additional analysis on the dendritic-cell stimulus-response dataset (Shalek et al., 2014). The dataset contains mouse bone-marrow-derived dendritic cells under LPS stimulation, including wild-type cells at baseline and 4 h after stimulation, together with experimentally observed 4 h knockout conditions for *Ifnar1*, *Stat1*, and *Tnfr*.

We learned couplings from wild-type baseline cells to wild-type 4 h LPS-stimulated cells and compared two in silico perturbation strategies for each pathway axis. In the gene-edit baseline, we zeroed out the perturbed gene in the wild-type endpoint snapshots before computing the predicted 4 h distribution. In the pathway-edit setting, we left the wild-type expression snapshots unchanged but modified the ligand–receptor catalogue so that the perturbation acted through the signalling-structure term in the coupling. We then evaluated each perturbation by comparing the predicted 4 h distribution against the experimentally observed 4 h knockout population.

*Table 17.* **In silico perturbation analysis on the dendritic-cell stimulus dataset.** We compare gene-level edits ($\alpha = 0$) and pathway-level edits ($\alpha = 1$) against experimentally observed knockout populations. Lower is better.

| Condition | Method | $W_1$ | $W_2$ |
|---|---|---|---|
| IFNAR1 KO | $\alpha = 0$ (gene edit) | 4.506 | 4.648 |
| IFNAR1 KO | $\alpha = 1$ (pathway edit) | **4.376** | **4.490** |
| STAT1 KO | $\alpha = 0$ (gene edit) | 5.606 | 5.686 |
| STAT1 KO | $\alpha = 1$ (pathway edit) | **5.502** | **5.567** |
| TNFR KO | $\alpha = 0$ (gene edit) | 4.487 | 4.572 |
| TNFR KO | $\alpha = 1$ (pathway edit) | **4.361** | **4.437** |

## G. Theoretical Analysis

### G.1. Synthetic setup

In this section, we provide a theoretical guarantee for the synthetic setup of Section C.1.

**Theorem 1.** *Let $\mathcal{D}_0$ and $\mathcal{D}_1$ be the source and target datasets defined by the synthetic clusters, and let $G^{(0)}, G^{(1)}$ be the associated directed interaction tensors.*

*Consider the two candidate couplings:*

1. *$\Gamma_{GT}$: The transport plan corresponding to the true translation vectors (preserving cluster identity).*

2. *$\Gamma_{FO}$: The transport plan corresponding to the feature-only map.*

*As in Section D.4, define the (unnormalized) feature and structure gaps between these two couplings as*

$$\Delta \mathcal{F} := \mathcal{F}(\Gamma_{FO}) - \mathcal{F}(\Gamma_{GT}), \qquad \Delta \mathcal{S} := \mathcal{S}(\Gamma_{FO}) - \mathcal{S}(\Gamma_{GT}),$$

*and the corresponding normalized feature and structure terms*

$$\tilde{\mathcal{F}}(\Gamma) := \frac{\mathcal{F}(\Gamma)}{|\Delta \mathcal{F}|}, \qquad \tilde{\mathcal{S}}(\Gamma) := \frac{\mathcal{S}(\Gamma)}{\Delta \mathcal{S}}.$$

*Let*

$$J(\Gamma, \alpha) = (1 - \alpha) \tilde{\mathcal{F}}(\Gamma) + \alpha \tilde{\mathcal{S}}(\Gamma)$$

*be the normalized FGW objective function.*

*Let $N$ denote the size of each cluster. Then, for sufficiently large $N$, there exists a critical threshold $\alpha^* \in (0, 1)$ such that for all $\alpha > \alpha^*$, the ground truth coupling strictly minimizes the objective relative to the feature-only alternative: $J(\Gamma_{GT}, \alpha) < J(\Gamma_{FO}, \alpha)$.*

*Proof.* Let the source measure be $\mu$ and the target measure be $\nu$, with the variance of the Normal distributions set to $\sigma^2 = 0.1$. The centroids are located at $\mu_0 = (-2, 2), \mu_1 = (0, 2), \mu_2 = (2, 2)$ and $\mu'_0 = (2, -2), \mu'_1 = (0, -2), \mu'_2 = (-2, -2)$. The interaction tensors $G^{(0)}$ and $G^{(1)}$ encode directed edges from the middle cluster ($k = 1$) to the left ($k = 0$) via Channel 1, and to the right ($k = 2$) via Channel 2. In what follows, we first compute the *unnormalized* feature and structure costs. We then incorporate the normalization scheme of Section D.4 in the final threshold derivation.

## 1. Analysis of the feature cost $\mathcal{F}$

For the ground truth coupling $\Gamma_{GT}$, each cluster $k$ maps to its true image $\mu'_k$. The cost is the mean squared norm of the translation vectors $v_0 = (4, -4)$, $v_1 = (0, -4)$, and $v_2 = (-4, -4)$:

$$\mathcal{F}(\Gamma_{GT}) = \frac{1}{3} \sum_{k=0}^{2} ||v_k||^2 = \frac{1}{3}(32 + 16 + 32) = \frac{80}{3}. \tag{22}$$

For the feature-only coupling $\Gamma_{FO}$, $\mathcal{S}_0$ maps to $\mathcal{S}'_2$, $\mathcal{S}_1$ to $\mathcal{S}'_1$, and $\mathcal{S}_2$ to $\mathcal{S}'_0$. In the finite sample regime with $N$ points, the optimal transport cost between two empirical Gaussian distributions with identical covariance matrices converges to the squared Euclidean distance between their means. We denote the finite-sample deviation by $\delta_N$:

$$\mathcal{F}(\Gamma_{FO}) = \frac{1}{3} \left( ||\mu_0 - \mu'_2||^2 + ||\mu_1 - \mu'_1||^2 + ||\mu_2 - \mu'_0||^2 \right) + \delta_N. \tag{23}$$

Hence:

$$\mathcal{F}(\Gamma_{FO}) = 16 + \delta_N. \tag{24}$$

The term $\delta_N$ represents the error between the empirical measures and their population counterparts. For distributions in dimension $d = 2$, this error decays at a rate of $\delta_N = O(N^{-1/2})$ (Fournier & Guillin, 2015). Provided $N$ is sufficiently large, standard OT ($\alpha = 0$) prefers the incorrect mapping since $16 < \frac{80}{3}$.

## 2. Analysis of structure cost $\mathcal{S}$

The structure cost is the Gromov-Wasserstein cost:

$$\mathcal{S}(\Gamma) = \sum_{i,k} \sum_{j,l} ||G^{(0)}_{ik} - G^{(1)}_{jl}||^2 \Gamma_{ij} \Gamma_{kl}. \tag{25}$$

Since $\Gamma_{GT}$ maps every source cluster $k$ to the target cluster with the same index $k$, and $G^{(1)}$ is defined to preserve the index-based structure of $G^{(0)}$, we have:

$$\mathcal{S}(\Gamma_{GT}) = 0. \tag{26}$$

For $\Gamma_{FO}$, the mapping permutes indices as $\pi(0) = 2, \pi(1) = 1, \pi(2) = 0$. We evaluate the cost for the two active interactions in $G^{(0)}$:

- **Edge** $1 \to 0$ **(Channel 1):** The source relation is $[1, 0]^\top$ and the target relation (from $\pi(1) = 1$ to $\pi(0) = 2$) is Channel 2 ($[0, 1]^\top$), which yields a squared difference of 2 with mass weight $1/9$.

- **Edge** $1 \to 2$ **(Channel 2):** The source relation is $[0, 1]^\top$ and the target relation (from $\pi(1) = 1$ to $\pi(2) = 0$) is Channel 1 ($[1, 0]^\top$), which yields a squared difference of 2 with mass weight $1/9$.

The total structure cost is:

$$\mathcal{S}(\Gamma_{FO}) = \frac{1}{9} \times 2 + \frac{1}{9} \times 2 = \frac{4}{9}. \tag{27}$$

### 3. Threshold derivation with normalization

We now incorporate the normalization scheme of Section D.4. Define the (unnormalized) feature and structure gaps between the two couplings as

$$\Delta\mathcal{F} := \mathcal{F}(\Gamma_{FO}) - \mathcal{F}(\Gamma_{GT}), \qquad \Delta\mathcal{S} := \mathcal{S}(\Gamma_{FO}) - \mathcal{S}(\Gamma_{GT}). \tag{28}$$

From the computations above,

$$\Delta\mathcal{F} = 16 + \delta_N - \frac{80}{3}, \qquad \Delta\mathcal{S} = \frac{4}{9}. \tag{29}$$

For sufficiently large $N$, we have $\mathcal{F}(\Gamma_{GT}) > \mathcal{F}(\Gamma_{FO})$, so $|\Delta\mathcal{F}| = \mathcal{F}(\Gamma_{GT}) - \mathcal{F}(\Gamma_{FO}) > 0$ and the normalization is well-defined. Normalizing, we get:

$$\tilde{\mathcal{F}}(\Gamma) = \frac{\mathcal{F}(\Gamma)}{|\Delta\mathcal{F}|}, \qquad \tilde{\mathcal{S}}(\Gamma) = \frac{\mathcal{S}(\Gamma)}{\Delta\mathcal{S}}. \tag{30}$$

The normalized FGW objective can therefore be written as

$$J(\Gamma, \alpha) = (1 - \alpha)\tilde{\mathcal{F}}(\Gamma) + \alpha\tilde{\mathcal{S}}(\Gamma). \tag{31}$$

For the two couplings of interest, we obtain

$$\tilde{\mathcal{S}}(\Gamma_{GT}) = \frac{\mathcal{S}(\Gamma_{GT})}{\Delta\mathcal{S}} = 0, \qquad \tilde{\mathcal{S}}(\Gamma_{FO}) = \frac{\mathcal{S}(\Gamma_{FO})}{\Delta\mathcal{S}} = 1, \tag{32}$$

and

$$\tilde{\mathcal{F}}(\Gamma_{GT}) - \tilde{\mathcal{F}}(\Gamma_{FO}) = \frac{\mathcal{F}(\Gamma_{GT}) - \mathcal{F}(\Gamma_{FO})}{|\Delta\mathcal{F}|} = \frac{|\Delta\mathcal{F}|}{|\Delta\mathcal{F}|} = 1. \tag{33}$$

We seek $\alpha$ such that $J(\Gamma_{GT}, \alpha) < J(\Gamma_{FO}, \alpha)$ under this normalization, i.e.

$$(1 - \alpha)\tilde{\mathcal{F}}(\Gamma_{GT}) < (1 - \alpha)\tilde{\mathcal{F}}(\Gamma_{FO}) + \alpha. \tag{34}$$

Using $\tilde{\mathcal{F}}(\Gamma_{GT}) - \tilde{\mathcal{F}}(\Gamma_{FO}) = 1$, this inequality becomes

$$(1 - \alpha) < \alpha \quad \Longleftrightarrow \quad \alpha > \frac{1}{2}. \tag{35}$$

Hence, under the normalization of Section D.4 and in the asymptotic regime, a critical threshold $\alpha^* = 1/2$ exists above which the ground truth coupling strictly improves the normalized objective relative to the feature-only alternative: $J(\Gamma_{GT}, \alpha) < J(\Gamma_{FO}, \alpha)$ for all $\alpha > 1/2$. $\qquad\square$

**Remarks.** In theory, $\alpha^* = 0.5$ comes from an idealized analysis of the normalized objective that only compares the feature-only and structure-only couplings in the *population limit*. Finite-sample effects, approximate normalization, and the existence of many 'almost-correct' couplings break the symmetry of the idealized setting and make the optimal $\alpha$ slightly bigger than 0.5.

Second, Theorem 1 shows that the ground truth coupling $\Gamma_{GT}$ is better than $\Gamma_{FO}$ at $\alpha = 1$. However, it is not the *only* one. The interaction tensors $G^{(0)}$ and $G^{(1)}$ are constant for all points within a cluster. Therefore, the structure cost $\mathcal{S}(\Gamma)$ depends only on which clusters are matched, not on how individual points are mapped within them.

Any coupling that correctly maps source clusters to their corresponding target clusters yields a structure cost of 0. This includes the ground truth coupling $\Gamma_{GT}$, but also any coupling that correctly matches clusters while randomly permuting points inside them. This explains the results observed in Figure 2: at $\alpha = 1$, the solver returns a solution that is structurally perfect but fails to recover the exact point-to-point correspondence.

## G.2. Dynamic interpretation of `CellBRIDGE`

We provide a dynamic viewpoint on `CellBRIDGE`, showing that it can be seen as the solution of a joint static-dynamic energy minimization problem combining kinetic energy in expression space and a structure-preserving term.

As before, let

$$\Pi(a, b) := \left\{ \Gamma \in \mathbb{R}_+^{n_0 \times n_1} : \Gamma \mathbf{1}_{n_1} = a, \ \Gamma^\top \mathbf{1}_{n_0} = b \right\}.$$

We further consider the common choice of feature cost

$$C_{ij} = \|x_i - y_j\|^2, \qquad 1 \le i \le n_0, \ 1 \le j \le n_1. \tag{36}$$

**Admissible processes for a fixed coupling.** Let $\Gamma \in \Pi(a, b)$ and define the associated joint law on endpoints

$$\Pi_\Gamma := \sum_{i=1}^{n_0} \sum_{j=1}^{n_1} \Gamma_{ij} \, \delta_{(x_i, y_j)}. \tag{37}$$

In the balanced case $\sum_{i,j} \Gamma_{ij} = 1$, so $\Pi_\Gamma$ is a probability measure with marginals $\rho_0, \rho_1$.

We consider continuous-time processes $(X_t)_{t \in [0,1]}$ taking values in $\mathbb{R}^d$ and satisfying:

- $X_\cdot$ has almost surely absolutely continuous paths

- the joint law of its endpoints is $(X_0, X_1) \sim \Pi_\Gamma$

We write $\mathcal{A}(\Pi_\Gamma)$ for the class of all such processes. For any $X_\cdot \in \mathcal{A}(\Pi_\Gamma)$, define the kinetic energy as:

$$\mathcal{K}(X_\cdot) := \mathbb{E}\left[ \int_0^1 \left\| \dot{X}_t \right\|^2 dt \right]. \tag{38}$$

The following lemma is standard but we include it for completeness.

**Lemma 1.** *Let $x, y \in \mathbb{R}^d$ and let $\gamma : [0, 1] \to \mathbb{R}^d$ be absolutely continuous with $\gamma(0) = x$, $\gamma(1) = y$. Then*

$$\int_0^1 \left\| \dot{\gamma}(t) \right\|^2 dt \ge \|y - x\|^2, \tag{39}$$

*with equality if and only if $\gamma(t) = (1 - t)x + ty$ for all $t \in [0, 1]$.*

*Proof.* By Cauchy–Schwarz inequality,

$$\left\| \int_0^1 \dot{\gamma}(t) \, dt \right\|^2 \le \int_0^1 \left\| \dot{\gamma}(t) \right\|^2 dt, \tag{40}$$

with equality if and only if $\dot{\gamma}(t)$ is constant in $t$. Since $\gamma(1) - \gamma(0) = y - x$, this yields

$$\|y - x\|^2 = \left\| \int_0^1 \dot{\gamma}(t) \, dt \right\|^2 \le \int_0^1 \left\| \dot{\gamma}(t) \right\|^2 dt, \tag{41}$$

Equality holds if and only if $\dot{\gamma}(t) = y - x$, i.e. $\gamma(t) = (1 - t)x + ty$. □

**Proposition 1.** *Let $\Gamma \in \Pi(a, b)$ and $\Pi_\Gamma$ be as above. Consider the admissible class $\mathcal{A}(\Pi_\Gamma)$ and the kinetic energy $\mathcal{K}$. Then:*

*1. The energy $\mathcal{K}(X_\cdot)$ is minimized over $\mathcal{A}(\Pi_\Gamma)$ by the process*

$$X_t^{\text{lin}} := (1 - t)X + tY, \qquad (X, Y) \sim \Pi_\Gamma. \tag{42}$$

2. *The minimal value of the kinetic energy is*

$$\inf_{X. \in \mathcal{A}(\Pi_\Gamma)} \mathcal{K}(X.) = \mathbb{E}_{(X,Y) \sim \Pi_\Gamma}\big[\|X - Y\|^2\big] = \sum_{i,j} \Gamma_{ij} C_{ij}. \tag{43}$$

*Proof.* Any $X. \in \mathcal{A}(\Pi_\Gamma)$ satisfies $(X_0, X_1) \sim \Pi_\Gamma$. Condition on the endpoints:

$$\mathcal{K}(X.) = \mathbb{E}_{(X,Y) \sim \Pi_\Gamma}\left[\mathbb{E}\Big[\int_0^1 \|\dot{X}_t\|^2 dt \,\Big|\, (X_0, X_1) = (X, Y)\Big]\right]. \tag{44}$$

For each fixed pair $(X, Y) = (x, y)$, Lemma 1 shows that the conditional energy is minimized by the straight-line path $t \mapsto (1-t)x + ty$, with minimal value $\|x - y\|^2$. Thus the global minimizer over $\mathcal{A}(\Pi_\Gamma)$ is the straight-line process $X_t^{\text{lin}}$, and

$$\inf_{X. \in \mathcal{A}(\Pi_\Gamma)} \mathcal{K}(X.) = \mathbb{E}_{(X,Y) \sim \Pi_\Gamma}\Big[\|X - Y\|^2\Big] = \sum_{i,j} \Gamma_{ij}\|x_i - y_j\|^2 = \sum_{i,j}\Gamma_{ij}C_{ij}. \tag{45}$$

$\square$

**Joint static–dynamic energy and reduction to FGW.** We can view `CellBRIDGE` as minimizing over both couplings and dynamics the joint energy functional

$$\mathcal{E}_\alpha(\Gamma, X.) := (1 - \alpha)\mathcal{K}(X.) + \alpha S(\Gamma), \tag{46}$$

subject to $\Gamma \in \Pi(a, b)$ and $X. \in \mathcal{A}(\Pi_\Gamma)$.

**Proposition 2.** *Fix $\alpha \in [0, 1]$. Consider the optimization problem*

$$\inf_{\Gamma \in \Pi(a,b)} \inf_{X. \in \mathcal{A}(\Pi_\Gamma)} \mathcal{E}_\alpha(\Gamma, X.) = \inf_{\Gamma \in \Pi(a,b)} \inf_{X. \in \mathcal{A}(\Pi_\Gamma)} \Big[(1 - \alpha)\mathcal{K}(X.) + \alpha S(\Gamma)\Big]. \tag{47}$$

*Then:*

1. *For any fixed $\Gamma$, the inner infimum over $X. \in \mathcal{A}(\Pi_\Gamma)$ is attained by the straight-line process $X_t^{\text{lin}} = (1-t)X + tY$, $(X, Y) \sim \Pi_\Gamma$, and*

$$\inf_{X. \in \mathcal{A}(\Pi_\Gamma)} \mathcal{E}_\alpha(\Gamma, X.) = (1 - \alpha)\sum_{i,j}\Gamma_{ij}C_{ij} + \alpha S(\Gamma). \tag{48}$$

2. *Consequently, the joint static–dynamic problem reduces to the purely static FGW problem*

$$\inf_{\Gamma \in \Pi(a,b)} \inf_{X. \in \mathcal{A}(\Pi_\Gamma)} \mathcal{E}_\alpha(\Gamma, X.) = \inf_{\Gamma \in \Pi(a,b)} \Big[(1 - \alpha)\langle \Gamma, C\rangle_F + \alpha S(\Gamma)\Big], \tag{49}$$

*whose minimizers are exactly the FGW-optimal couplings used by `CellBRIDGE`.*

*Proof.* Point (1) follows directly from Proposition 1: for any $\Gamma$,

$$\inf_{X. \in \mathcal{A}(\Pi_\Gamma)} \mathcal{E}_\alpha(\Gamma, X.) = (1 - \alpha)\inf_{X. \in \mathcal{A}(\Pi_\Gamma)} \mathcal{K}(X.) + \alpha S(\Gamma) = (1 - \alpha)\sum_{i,j}\Gamma_{ij}C_{ij} + \alpha S(\Gamma). \tag{50}$$

Taking the infimum over $\Gamma \in \Pi(a, b)$ yields (2), which coincides with the static FGW objective. $\square$

Intuitively, Proposition 2 shows that `CellBRIDGE` does not use linear interpolations between matched cells as a heuristic, but as the *unique* minimal-action choice once the coupling $\Gamma^\star$ is fixed. The static FGW step therefore selects an interaction-aware coupling that trades off feature displacement and CCI preservation, and the subsequent dynamic step realizes this coupling by approximating the lowest-kinetic-energy flow in expression space. When $\alpha = 0$, this recovers the classical OT–CFM (Tong et al., 2024a)/ Benamou-Brenier (Benamou & Brenier, 2000) interpolation.

### G.3. Connection to the velocity field learned with CFM.

In practice, `CellBRIDGE` does not explicitly construct the process $X_t^{\text{lin}}$ but instead uses CFM to learn a time–dependent vector field $v_\theta$ that generates the same probability path.

In the infinite–capacity and optimization limit, the minimizer $v^\star$ of $\mathcal{L}_{\text{CFM}}$ coincides with the velocity field of $X_t^{\text{lin}}$ constructed in Section G.2, in the sense that

$$v^\star(z, t) = \mathbb{E}[Y - X \mid Z_t = z],$$

and the ODE

$$\dot{z}_t = v^\star(z_t, t)$$

generates exactly the probability path $\{\rho_t\}_{t \in [0,1]}$ induced by $\Gamma^\star$.

We can relate $v^\star$ to the kinetic energy of the straight–line process, following a similar technique as in (Lipman et al., 2024). For a time–dependent vector field $w : \mathbb{R}^d \times [0, 1] \to \mathbb{R}^d$ that generates $\{\rho_t\}$, define its kinetic energy along this path by

$$\mathcal{K}_{\text{Eul}}(w) := \int_0^1 \mathbb{E}_{Z_t \sim \rho_t}\Big[ \|w(Z_t, t)\|_2^2 \Big] dt.$$

Using the formula of $v^\star$ above and Jensen's inequality, we obtain

$$\begin{aligned}
\mathcal{K}_{\text{Eul}}(v^\star) &= \int_0^1 \mathbb{E}\Big[ \big\|\mathbb{E}[Y - X \mid Z_t]\big\|_2^2 \Big] dt \\
&\leq \int_0^1 \mathbb{E}\Big[ \mathbb{E}\big[\|Y - X\|_2^2 \mid Z_t\big] \Big] dt \\
&= \int_0^1 \mathbb{E}\Big[ \|Y - X\|_2^2 \Big] dt \\
&= \mathbb{E}_{(X,Y) \sim \Pi}\Big[ \|Y - X\|_2^2 \Big].
\end{aligned}$$

By Proposition 1 we have

$$\mathbb{E}_{(X,Y) \sim \Pi_{\Gamma^\star}}\Big[ \|Y - X\|_2^2 \Big] = \sum_{i,j} \Gamma_{ij}^\star C_{ij} = K\big(X_\cdot^{\text{lin}}\big),$$

Hence

$$\mathcal{K}_{\text{Eul}}(v^\star) \leq \sum_{i,j} \Gamma_{ij}^\star C_{ij} = K\big(X_\cdot^{\text{lin}}\big).$$

In other words, for a fixed coupling $\Gamma$, the feature term

$$F(\Gamma) = \langle \Gamma, C \rangle_F = \sum_{i,j} \Gamma_{ij} C_{ij}$$

provides an explicit upper bound on the kinetic energy of the velocity field recovered by CFM from the corresponding straight–line dynamics. Combined with the joint static–dynamic formulation in Equation (46), this shows that the `CellBRIDGE` objective

$$(1 - \alpha) \langle \Gamma, C \rangle_F + \alpha\, S(\Gamma)$$

can be viewed as selecting a coupling that balances CCI preservation with a surrogate upper bound on the kinetic energy of the flow that CFM learns from that coupling.

