# OpenReview forum: "CellBRIDGE: Learning Cellular Trajectories via Interaction-Aware Alignment"
_ICML.cc/2026/Conference — ICML 2026 regular_

### Official Review · Reviewer_XWAm · 2026-03-11

**Soundness:** 3
**Presentation:** 3
**Significance:** 2
**Originality:** 2
**Overall Recommendation:** 4
**Confidence:** 4

**Summary:**

The paper shows that incorporating intra-snapshot structure, through cell–cell communication mediated by ligand–receptor signaling, improves cellular trajectory inference between individual snapshots of single cell measurements.

This is shown by comparing the Wasserstein distance between the predicted gene expression at time t_1, given t_0 and t_2, and the true t_1, with and without the interaction-Aware Alignment. What is interesting about this method is that the CCI intra-snapshot structure can be incorporated into existing methods, and the authors show that doing so yields better results across several methods.

**Compliance With Llm Reviewing Policy:**

Affirmed.

**Final Justification:**

The authors have addressed my main concerns and reinforced my positive assessment of the paper. The reason why the score is not higher is that the overall impact is limited (for this venue), but the work is original, the authors do a great job at presenting it and their empirical section is well constructed and comprehensive.

**Key Questions For Authors:**

No further questions at this stage.

**Limitations:**

yes

**Strengths And Weaknesses:**

The submission is technically sound. The authors do a good job at showing that the interaction-aware alignment improves trajectory inference, by replacing the default coupling of several popular methods with the interaction-aware coupling produced by CellBRIDGE. Additionally, they show through an ablation study that the improvement is not due to arbitrary regularization but by the LR structure.

The paper is well written and easy to follow. The part that I think can be improved is the one on related work, where it feels like a list of methods without giving much intuition about how they work and what makes them different among each other. Also, this should be a new paragraph: Dynamic priors: directionality and population-level constraints.

The idea of using ligand receptor pairs to bind together consecutive snapshot of single cell data is clever and original. It points to a growing evidence in single cell biology, whereas leveraging known information about the problem at hand can help computational models perform better across tasks. (here is an example of a loss function that leverages the structure of the ontology for cell annotation: https://www.nature.com/articles/s43588-025-00945-z , or prior knowledge of gene perturbations for genetic interventions: https://www.biorxiv.org/content/10.1101/2025.06.27.661992v1.full.pdf). These points on integration of prior knowledge in learning frameworks could be included in the discussion.

When figure 1 is introduced, the reader does not know what G_ij is, which makes the figure message hard to grasp and the caption does not help. Also the aesthetic of figure 1 can be improved.

The main weakness is that the overall impact of the problem studied in the paper is likely limited: holding out a temporal snapshot and marginally improving existing methods at reconstructing it does not on its own unlock new biological discoveries nor opens up avenues for further machine learning research follow ups.

---

> ### Author Rebuttal · Authors · 2026-03-31
>
> Thank you for the positive assessment of our work. We address your comments below.
>
> ---
>
> ### [P1] Significance of CellBRIDGE
>
> We appreciate your concern regarding the impact of the studied problem and would like to address this from both the machine learning and biological perspectives.
>
>
> **Trajectory inference is important.** We agree that a small gain in held-out interpolation, by itself, would have limited impact. However, this is not the claim of our paper. Rather, interpolation is our evaluation proxy for a hard and biologically important problem: inferring cellular dynamics from destructive single-cell snapshots. This problem is central in biology because understanding how cell populations evolve over time underlies development, disease progression, and therapeutic response. The impact of faithful trajectory inference is therefore better mechanistic understanding and hypothesis generation in settings where direct longitudinal tracking of single cells is impossible.
>
> **New cost function.**  Our work addresses a broader open question in transport-based trajectory inference: which biological priors should define the transport cost? This (partially) answers the call of [1]: *"the adaptive selection of robust cost functions remains a crucial area for future research"*. Existing OT methods mostly rely on gene-expression distances. In contrast, CellBRIDGE shows that directed, typed ligand–receptor communication can be injected in the coupling level to improve downstream trajectory inference. We view this as a methodological contribution on cost design for biological OT, which directly opens follow-up work on how to integrate more mechanistic transport costs in biology.
>
> **Biological relevance.** On the biology side, the main benefit is not only improved interpolation, but that CellBRIDGE enables pathway-level in silico perturbation analyses not available in feature-only transport formulations. In `Section 5.3`, we simulate such perturbations by ablating specific ligand–receptor pathways from the interaction prior. On the lung tumor dataset, attenuating EGFR-, ALK-, or MET-related signalling produces measurable decreases in tumor-associated progression scores, whereas unrelated control pathways have negligible effects. These results suggest that the framework can support biologically informed hypothesis generation about cell–cell communication, beyond just improved snapshot reconstruction.
>
> **Summary:** We believe the impact is twofold:  1) methodologically, we introduce a class of interaction-aware cost functions for trajectory inference; 2) biologically, this enables perturbation analysis of intercellular signalling at the trajectory level.
>
> [1] Bunne et al., Optimal transport for single-cell and spatial omics
>
> ### [P2] Related work
> **Better organization.** We agree that the related-work section would benefit from a more structured comparison. To make this more clear, we have moved our `Table 3`(which is currently in the appendix) to the main text so that readers can have more intuition on the differences between the different methods.
>
> **Other references.** We also agree that the manuscript would benefit from a broader discussion of related methods that incorporate biological inductive biases in single-cell analysis. In addition to the references you suggested, we now also discuss work that injects pathway knowledge and cell ontology structure for cell-type annotation [2,3], as well as biologically informed Neural ODEs for regulatory dynamics [4]. These works further support the broader view that structured biological prior knowledge can improve single-cell learning, while CellBRIDGE contributes a complementary inductive bias at the level of ligand–receptor communication structure.
>
>
>
> [2] Tang et al., Knowledge-based inductive bias and domain adaptation for cell type annotation (Communication Biology 2025)
>
> [3] Wang et al., Leveraging the Cell Ontology to classify unseen cell types (Nature Communications 2021)
>
> [4] Hossain et al., Biologically informed Neural ODEs for genome-wide regulatory dynamics (Genome Biol. 2024)
>
> ### [P3] Presentation
> Thank you for pointing out that `Fig. 1`can be improved. We have revised it with an informative legend and now explicitly define $G_{ij}$ and $D$ in the caption.
>
> ---
>
> Thank you again for the detailed feedback. We hope our responses have fully addressed your comments, and we would be glad to clarify any remaining points.

---

> > ### Author Rebuttal · Reviewer_XWAm · 2026-04-01
> >
> > I thank the authors for their careful response. I am glad we are aligned on [P2] and [P3]. The pathway-level in silico perturbation analysis, and the corresponding ablation study, is indeed my favorite part of the paper. It would be great to see it validated beyond the Lung Tumor dataset. I will keep my score, all points have been addressed.

---

> > > ### Author Response · Authors · 2026-04-07
> > >
> > > Thank you for the careful follow-up, we are glad that our responses resolved the concerns and that you recommend acceptance.
> > >
> > > ---
> > >
> > > We agree with you that the ability to do *in silico* perturbation analysis via the CCI tensors is a key strength of the paper.
> > > Following your suggestion, we performed a new analysis beyond the Lung Tumor dataset on the independent dendritic stimulus dataset from Shalek et al. [1].
> > >
> > > **Dataset description.** The dataset contains mouse bone-marrow-derived dendritic cells under LPS stimulation, with WT cells at 0 h and 4 h, together with experimentally observed 4 h knockout conditions for `Ifnar1`, `Stat1`, and `Tnfr` [1].
> > >
> > > **Setup.** We learned couplings from WT 0 h to WT 4 h and considered two *in silico* perturbation strategies for each pathway axis:
> > > (i) **gene edit** ($\alpha = 0$), where we zero out the perturbed gene in the WT endpoint snapshots
> > > (ii) **pathway edit** ($\alpha = 1$), where we don't modify the WT snapshots but modify the ligand–receptor catalog so that the coupling is driven by the signaling-structure term.
> > >
> > > We then evaluated each perturbation by comparing the predicted pushed 4 h distribution against the experimentally observed 4 h knockout population.
> > >
> > >
> > > **Results.** We report the results in the following table:
> > >
> > >
> > > | Condition | Method | $W_1$ | $W_2$ |
> > > |---|---|---:|---:|
> > > | `IFNAR1 KO` | $\alpha = 0$ (gene edit)  | 4.506 | 4.648 |
> > > | `IFNAR1 KO` | $\alpha = 1$ (pathway edit) | 4.376 | 4.490 |
> > > | `STAT1 KO` | $\alpha = 0$ (gene edit) | 5.606 | 5.686 |
> > > | `STAT1 KO` | $\alpha = 1$ (pathway edit) | 5.502 | 5.567 |
> > > | `TNFR KO` | $\alpha = 0$ (gene edit) | 4.487 | 4.572 |
> > > | `TNFR KO` | $\alpha = 1$ (pathway edit) | 4.361 | 4.437 |
> > >
> > > Across all three perturbation axes, the $\alpha = 1$ pathway edit achieves lower $W_1$ and $W_2$ than the $\alpha = 0$ gene-edit baseline. Thus, on this independent dataset, directly editing signaling structure yields a more faithful approximation of the observed perturbed population than zeroing out a single gene alone. This suggests that the perturbation is better captured at the *pathway / interaction level* than by zeroing out a single gene in isolation.
> > >
> > >
> > > [1] *Single-cell RNA-seq reveals dynamic paracrine control of cellular variation* (Shalek et al., *Nature*, 2014)
> > >
> > > ---
> > >
> > > Thank you again for the constructive feedback and thoughtful engagement! If you feel that this additional validation on an independent immune dataset strengthens the paper, we would be grateful if you would take it into account in your final assessment.

---

### Official Review · Reviewer_ueZ5 · 2026-03-15

**Soundness:** 3
**Presentation:** 3
**Significance:** 2
**Originality:** 3
**Overall Recommendation:** 4
**Confidence:** 2

**Summary:**

This paper introduces an approach for modeling cellular trajectory via optimal transport. The core insight of the method lies in the development of a principled OT framework, including a biologically inspired OT cost design and a corresponding transport plan assignment . Empirical results demonstrate the superiority of the proposed approach in cellular trajectory modeling.

**Compliance With Llm Reviewing Policy:**

Affirmed.

**Final Justification:**

The concerns are addressed and I maintain the positive score. However, I am not an expert in this area and my weight on the justification should be downgraded.

**Key Questions For Authors:**

1. How does the OT-based approaches compare with the kernel-based distances (e.g., used in MMD) which could play a similar role as the transport plan?

2. The results in Table 1 seem to suggest that OT-CFM, OT-MFM, and UOT-FM can already achieve similar performance compared with the proposed approach. In this case, how significant is the improvement?

**Limitations:**

Yes.

**Strengths And Weaknesses:**

## Strengths:

1. The paper is well motivated and the approach is designed in detailed and careful manner with biological inspirations.

2. The method demonstrates strong performance in modeling cellular trajectories on several benchmarks again a wide suite of baselines.

3. The paper has clear presentation and the method appears to be reasonable under the context of optimal transport.

## Weaknesses

1. The improvement in performance of the proposed approach compared with some baselines, e.g., OT-CFM, seem to be marginal.

2. The scalability of the approach can be verified by experiments of larger scales.

---

> ### Author Rebuttal · Authors · 2026-03-31
>
> Thank you for your positive assessment. We address your comments below.
>
> ---
>
>
>
> ### [P1] Significance of improvements
> **Statistical significance.** We appreciate your concern regarding the significance of the results. To address this point, we have conducted statistical tests to ensure that the gains are significant and unlikely to be explained by noise. Across 35 paired comparisons, CellBRIDGE with $\alpha=0.5$ yields consistent improvements over feature-OT baselines across all model families (see our response to `Reviewer QvcQ`). Despite the small absolute magnitude of improvements, these results show that the gains are statistically significant, consistent across settings, and accompanied by small-to-moderate effect sizes.
>
> **Trajectory-level analysis.** Moving beyond distributional metrics ($W_1$ and $W_2$), we notice that incorporating structural information changes the inferred coupling in a biologically meaningful way at the *level of individual trajectories*. We show this in `Section 5.3`, where CellBRIDGE enables mechanism-level counterfactual analysis: editing the ligand–receptor prior shifts the inferred trajectories in a pathway-specific manner, with EGFR, ALK, and MET ablations producing measurable decreases in tumor-associated progression scores, while unrelated control pathways have negligible effect. The key value of the method is therefore not only improved distributional metrics, but the ability to produce editable and interaction-aware trajectories that can be analyzed.
>
> **Impact.** More generally, we hope that our work can inspire future research into an open problem explicitly identified and mentioned by [1], i.e. *"the adaptive selection of robust cost functions remains a crucial area for future research".*
>
> [1] Bunne, Charlotte, et al. "Optimal transport for single-cell and spatial omics."
>
> ### [P2] Scalability of experiments
> Following your suggestion, we extended the evaluation to substantially larger datasets with two additional interpolation tasks on a mouse development cell atlas used in the original moscot paper [2]. These results show that CellBRIDGE remains feasible in this larger-scale regime.
>
> We did not include UOT-FM on this benchmark because the unbalanced OT solver did not scale to datasets of this size in our experiments.
>
> | Method            |                 E7.5→E8 $W_1/W_2$ |             E7.75→E8.25 $W_1/W_2$ |
> | ----------------- | --------------------------------: | --------------------------------: |
> | OT-CFM            |     3.266 ± 0.011 / 3.575 ± 0.023 |     3.229 ± 0.018 / 3.249 ± 0.010 |
> | OT-MFM            |     3.249 ± 0.007 / 3.586 ± 0.025 |     3.235 ± 0.015 / 3.250 ± 0.007 |
> | SF2M              |     4.967 ± 0.078 / 5.382 ± 0.078 |     4.748 ± 0.108 / 4.884 ± 0.125 |
> | CellBRIDGE + CFM  |     3.068 ± 0.007 / 3.555 ± 0.005 |     3.131 ± 0.022 / 3.323 ± 0.037 |
> | CellBRIDGE + MFM  |     3.054 ± 0.009 / 3.546 ± 0.014 |     3.140 ± 0.023 / 3.384 ± 0.020 |
> | CellBRIDGE + SF2M |     4.585 ± 0.110 / 4.872 ± 0.132 |     4.767 ± 0.238 / 4.947 ± 0.225 |
>
>
> [2] Klein et al., Mapping cells through time and space with moscot (Nature 2025)
>
> ### [P3] Comparison to kernel-based distances
>
> **Conceptual difference.** OT and kernel-based distances such as MMD are conceptually different in our setting. Our method uses OT to construct a coupling between the two snapshots, which in turn defines an interpolating marginal path. By contrast, MMD is primarily a discrepancy measure between distributions: it can quantify how close two snapshots are, but it does not by itself provide a sample-level coupling or an interpolation mechanism of the kind used in our framework.
>
> **Empirical comparison.** That said, we agree that a kernel-based baseline is informative. We therefore compared against SnapMMD [3], which trains a velocity field using an MMD-based objective. Across all datasets, OT-based methods consistently outperform SnapMMD.
> | Method                          |            V1 Light $W_1/W_2$ |  Dendritic Stimulus $W_1/W_2$ |          Lung tumor $W_1/W_2$ |
> | ------------------------------- | ----------------------------: | ----------------------------: | ----------------------------: |
> | SnapMMD                         | 2.420 ± 0.004 / 2.657 ± 0.004 | 3.863 ± 0.032 / 4.022 ± 0.043 | 2.237 ± 0.128 / 2.520 ± 0.103 |
> | CellBRIDGE + CFM ($\alpha=0.5$) | 2.381 ± 0.004 / 2.618 ± 0.003 | 3.679 ± 0.009 / 3.835 ± 0.010 | 1.989 ± 0.004 / 2.272 ± 0.005 |
>
>
> [3] Berlinghieri et al., Oh SnapMMD! Forecasting Stochastic Dynamics Beyond the Schrödinger Bridge’s End.
>
> ---
>
> Thank you again for the detailed feedback. We hope our responses have fully addressed your comments, and we would be glad to clarify any remaining points.

---

> > ### Author Rebuttal · Reviewer_ueZ5 · 2026-04-03
> >
> > The concerns are addressed and I maintain my positive score.

---

> > > ### Author Response · Authors · 2026-04-07
> > >
> > > Thank you for your response, we are glad that your questions have been fully resolved, and want to thank you for your constructive feedback in this review process!

---

### Official Review · Reviewer_QvcQ · 2026-03-16

**Soundness:** 2
**Presentation:** 2
**Significance:** 2
**Originality:** 2
**Overall Recommendation:** 4
**Confidence:** 2

**Summary:**

The authors propose a framework that augments feature-based OT with a directed cell-cell interaction cost derived based on ligand-receptor activity.

**Compliance With Llm Reviewing Policy:**

Affirmed.

**Final Justification:**

The authors have addressed my concerns.

**Key Questions For Authors:**

The differences in performance seem quite small (Tables 1-2). Could you run some statistical tests to determine whether the performance gains are significant? For Table 2, could you run the analyses on multiple seeds?

For Figure 5b: Could you clarify your claim about how, in contrast to OT, the CellBridge is not susceptible to the assumption that "snapshots are separated by modest temporal gaps and the system evolves smoothly"? It seems like the CCIs are not helpful, which suggests to me that CellBridge suffers from the same problem as OT.

**Limitations:**

Yes

**Strengths And Weaknesses:**

Strengths
- The paper is well-motivated and generally clear
- The approach is intuitive

Weaknesses
- See questions below

---

> ### Author Rebuttal · Authors · 2026-03-31
>
> Thank you for your constructive review. We address your comments below.
>
> ---
>
> ### [P1] Significance of results
>
> **Statistical tests.** Following your suggestion, we have performed statistical tests to quantify the significance of the results. Across 35 paired comparisons, CellBRIDGE with $\alpha=0.5$ yields consistent improvements over feature-OT baselines across all model families.
>
> | Method              | Metric | Rel. imp. (%) | Cohen’s $d_z$ | Wilcoxon $p$ | Holm-adj. $p$ |
> | --------- | ------ | ------------: | ------------: | -----------: | ------------: |
> | CellBRIDGE + CFM    | $W_1$  |          0.56 |          0.63 |      1.53e-4 |       9.19e-4 |
> |                     | $W_2$  |          0.58 |          0.63 |      1.91e-4 |       1.53e-3 |
> | CellBRIDGE + MFM    | $W_1$  |          0.31 |          0.44 |      1.60e-3 |       6.38e-3 |
> |                     | $W_2$  |          0.34 |          0.44 |      4.12e-3 |       2.88e-2 |
> | CellBRIDGE + SF2M   | $W_1$  |          0.89 |          0.29 |      8.30e-3 |       2.49e-2 |
> |                     | $W_2$  |          0.84 |          0.27 |      9.12e-3 |       4.56e-2 |
> | CellBRIDGE + UOT-FM | $W_1$  |          0.89 |          0.48 |      1.31e-4 |       9.19e-4 |
> |                     | $W_2$  |          0.71 |          0.34 |      4.34e-3 |       2.88e-2 |
>
>
> While absolute improvements are modest, the consistency across datasets, models, and metrics indicates a systematic improvement in coupling quality. We also added two new larger datasets with stronger improvements during the rebuttal (see our response to `Reviewer ueZ5`).
>
> **Table 2 with multiple seeds**. The CellBRIDGE results reported in `Table 2` are deterministic, as it uses full snapshots and the exact interpolation defined by the coupling and affine interpolation, with no stochastic component. Therefore, the only methods in `Table 2` affected by randomness are  Shuffle, Random LR and Metacell. We have rerun these baselines over 5 random seeds:
>
> | Method         | Tumor $W_1$     | Tumor $W_2$     | Dendritic $W_1$ | Dendritic $W_2$ | Light $W_1$     | Light $W_2$     |
> | -------------- | --------------- | --------------- | --------------- | --------------- | --------------- | --------------- |
> | *Shuffle*      | 2.188 ± 0.007   | 2.392 ± 0.007   | 3.644 ± 0.004   | 3.748 ± 0.007   | 2.441 ± 0.002   | 2.646 ± 0.002   |
> | *Random LR*    | 2.171 ± 0.030   | 2.388 ± 0.033   | 3.602 ± 0.022   | 3.729 ± 0.019   | 2.446 ± 0.006   | 2.655 ± 0.008   |
> | *Metacell*     | 2.052 ± 0.002   | 2.344 ± 0.002   | 3.587 ± 0.005   | 3.725 ± 0.000   | 2.329 ± 0.000   | 2.567 ± 0.002   |
> | **CellBRIDGE** | 2.028           | 2.298           | 3.585           | 3.732           | 2.350           | 2.587           |
>
>
>
>
>
> **Trajectory-level analysis.** Moving beyond distributional metrics, we notice that incorporating structural information changes the inferred coupling in a biologically meaningful way at the *level of individual trajectories*. We show this in `Section 5.3`, where CellBRIDGE enables mechanism-level counterfactual analysis: editing the ligand–receptor prior shifts the inferred trajectories in a pathway-specific manner. The key value of the method is therefore not only improved distributional metrics, but the ability to produce editable and interaction-aware trajectories that can be analyzed.
>
> **Impact.** More generally, we hope that our work can inspire future research into an open problem explicitly identified and mentioned by Bunne et al., i.e. *"the adaptive selection of robust cost functions remains a crucial area for future research".*
>
>  Bunne, Charlotte, et al. "Optimal transport for single-cell and spatial omics."
>
>
>
> ### [P2] Figure 5b clarification
>
> **Smoothness assumption.** Our intent in `Fig. 5b` is to highlight exactly the limitation you mentioned. CellBRIDGE, like OT-based alignment more broadly, assumes that nearby snapshots are meaningfully alignable over modest temporal gaps. In the embryo experiment, this assumption breaks down: rapid remodeling over a six-day interval makes cross-snapshot CCI structure non-transferable, so increasing $\alpha$ does not help.
>
> **Difference with classic OT.** We want to clarify that we do *not* claim that CellBRIDGE removes the smoothness assumption. Rather, it introduces an orthogonal prior to *feature-only OT* in order to better constrain underdetermined alignments. Consistent with this, `Sections 5.1` and `5.2` show improvements over $\alpha = 0$, and plugging CellBRIDGE into SF2M, MFM, and UOT-FM also yields gains, indicating that the benefit arises from the coupling-level biological prior rather than a specific downstream model.
>
> ---
>
> Thank you again for the detailed feedback. We hope our responses have fully addressed your comments, and we would be glad to clarify any remaining points.

---

> > ### Author Rebuttal · Reviewer_QvcQ · 2026-04-04
> >
> > Thank you for your efforts! I will raise my score +1.

---

> > > ### Author Response · Authors · 2026-04-07
> > >
> > > Thank you for your response, we are glad that your questions have been fully resolved, and that you recommend acceptance! We also want to thank you for your constructive feedback in this review process.

---

### Official Review · Reviewer_1LJW · 2026-03-22

**Soundness:** 3
**Presentation:** 3
**Significance:** 2
**Originality:** 2
**Overall Recommendation:** 4
**Confidence:** 4

**Summary:**

The manuscript focuses on inferring dynamics from population snapshots, with applications to single-cell RNA sequencing. In particular, it studies the ground cost in an optimal transport formulation used to infer dynamics between evolving distributions. The method leverages cell–cell interactions paired with fused Gromov–Wasserstein problem.

**Compliance With Llm Reviewing Policy:**

Affirmed.

**Key Questions For Authors:**

Minor comments and questions:

- p3 l161: Please add a citation for the Sinkhorn algorithm.
- p5 l267: The authors should add citations for each subsection like MFM, SF2M, etc.
- p13 l675: References should be reviewed. Some entries are duplicated, e.g., *Vayer, T., Chapel, L., Flamary, R., Tavenard, R., and Courty, N. Fused Gromov-Wasserstein distance for structured objects.*
- p5 l225: $g^{(.)}_{i\rightarrow k}$ is already defined; is it the same as in line 212?
- p5 l251: Does this mean that, to find the optimal value for $\alpha = 0.5$, one needs to solve the problem for $\alpha = 0$, $\alpha = 1$, and $\alpha = 0.5$? Are there alternative normalization strategies the authors could consider that do not require multiple optimization steps?
- In figure 3, how should the pathway information for A and B be interpreted? I understand that the color encodes the CCI and that the goal is to find a pairing that preserves it. Do the colors also correspond to specific pathways?
- p7 l341: In practice, mini-batches are used to train these flow-based models. Is the proposed method's coupling matrix computed on mini-batches or on the full dataset? If computed on the full dataset, how does it scale? Since GW can be computationally expensive, did the authors encounter any practical limitations? It would be helpful to include a discussion of this in the manuscript (other than in the limitation). If mini-batches are used, can the authors quantify the accuracy of the approximation?
- p8 l418: The authors highlight a limitation related to non-persistent signaling patterns. In practice, is it possible to detect changes in signaling patterns from the data? Can a practitioner determine whether the method is appropriate for a new dataset?
- p5 l227: typo with $\Pi$.

**Limitations:**

yes

**Strengths And Weaknesses:**

While the paper provides a solid and careful empirical study, the overall practical impact and performance gains of the proposed method are limited. The normalization, which requires solving three optimization problems, seems like a major limitation of the method. Additionally, if the authors use mini-batches to approximate the coupling, they should report how sensitive this approximation is to the batch size.

The experimental section is well structured and focuses on the main components of the method: the coupling and its use in downstream tasks. Although the improvements reported (e.g., in Table 1) are modest, the results are consistent and the comparisons are of high quality. The limitations section is particularly valuable, as it helps the reader understand for which tasks or datasets the method is best suited. Figure 5b clearly highlights some limitations while guiding users toward choosing a smaller $\alpha$ as a safer parameter setting. Overall, even though the method does not yield major improvements, the experimental evaluation is thorough and well executed: the authors compare against multiple baselines, provide ablations and toy datasets to illustrate the benefits of the method, and explicitly expose limitations through experiments. The paper contributes to improving the understanding of the problem.

---

> ### Author Rebuttal · Authors · 2026-03-31
>
> Thank you for your positive and constructive review. We address your comments below.
>
> ---
>
> ### [P1] Impact
> While gains in interpolation metrics are modest (though statistically significant; see response to `Reviewer QvcQ`), the contribution extends beyond reconstruction accuracy. CellBRIDGE introduces a new inductive bias at the level of the OT coupling that changes inferred cell-level correspondences and downstream trajectories, enabling counterfactual analysis over cell–cell communication in a way that feature-only alignment methods do not (see our response to `Reviewer XWAm`). We therefore view the impact primarily in terms of new modeling capability rather than incremental gains in standard metrics. We also strengthened the empirical evidence by adding two larger datasets, where the improvements are more pronounced (see our response to `Reviewer ueZ5`).
>
> ### [P2] Normalization
> You are correct that our normalization requires solving two OT problems (for $\alpha=0$ and $\alpha=1$) to calibrate the relative scales of the cost and structural terms. The motivation is that $\alpha$ should smoothly control the balance between the two terms.
> We tested a simpler normalization strategy on the tumour dataset that avoids these endpoint OT solves, by scaling $C$, $G^{(0)}$, and $G^{(1)}$ by their respective medians. These results suggest that our normalization provides better calibration between the feature and structural terms.
>
> | $\alpha$| OT Norm. W1|Median Norm. W1| OT Norm. W2|Median Norm W2 |
> |--:|---:|--:|--:|--:|
> | 0.0 | 1.934 | 1.934| 2.214| 2.214|
> | 0.1 | 1.929  | 1.919|2.210| 2.203|
> | 0.2 | 1.924 | 1.921|2.206| 2.204|
> | 0.3 | 1.919 | 1.932|2.204| 2.212|
> | 0.4 | 1.914| 1.942 | 2.200| 2.221|
> | 0.5 | 1.919| 1.955| 2.203| 2.231|
> | 0.6 | 1.919| 1.970| 2.203| 2.242|
> | 0.7 | 1.921| 1.983 | 2.204| 2.253|
> | 0.8 | 1.933| 1.998| 2.214| 2.268|
> | 0.9 | 1.956| 2.012| 2.231| 2.281|
> | 1.0 | 2.028 | 2.028| 2.298| 2.298|
>
> ### [P3] Computational cost
> **Coupling.** In all experiments, the coupling is computed once at the *full snapshot level*, i.e. using the full source and target snapshots rather than mini-batches. Mini-batching is used only when we train the downstream flow model, and we sample from the joint distribution induced by the coupling to define the batches. We will clarify this in `Sec. 4.2`.
>
> **Scalability.** We agree that the FGW objective can be computationally expensive. The cost of our custom conditional-gradient solver is dependent on the underlying OT solver for the linear OT subproblems. We provide a discussion of this cost in `Appendix D.9`. Empirically, the linear programming solver from the POT's library did not create a practical bottleneck for the datasets we considered, i.e.  the OT computation typically ran in under 5 minutes. We also added experiments on a larger mouse embryo cell atlas to demonstrate the scalability of our method (see response to `Reviewer ueZ5`).
>
> **Possible approximations.**  `Appendix D.9`  discusses standard scalability strategies that are orthogonal to our formulation, including entropic regularization / Sinkhorn-type solvers instead of LP solvers, mini-batch OT as you suggested, and metacell aggregation to reduce sample size.
>
> ### [P4] When does the structural prior help?
> In practice, when at least three snapshots are available, the utility of incorporating structure can be assessed directly by holding out the middle snapshot to tune $\alpha$. With only two snapshots, this is harder to quantify. In that case, the method is best suited to modest temporal gaps and relatively smooth evolution, which can be estimated via the GW cost between snapshots. Orthogonal annotations (e.g. cell types, lineage labels, or marker programs) can also help assess whether $\alpha>0$ yields biologically plausible couplings.
>
> ### [P5] Figure 3
> We apologize for the lack of clarity in Fig. 3 and will clarify this in the main text. The pathways define a two-channel interaction structure. One channel encodes directed interactions from orange to blue points, and the other channel encodes interactions from orange to green points. Therefore, the CCIs $G^{(0)}$ and $G^{(1)}$ lie in $\mathbb{R}^{n \times n \times 2}$. This interaction structure explains why $\alpha=0$ fails: with the feature-only OT coupling, pairs of (orange, blue) are mapped to pairs of (orange, green).
>
> ### [P6] Typos and references
> We apologize for the confusing notation. $g^{(0)}\_{i\to k}$ denotes the vector of interaction between $i$ and $k$ (hence a vector in $\mathbb{R}^{P}$), whereas at line 212, $q^{(p)}\_{i\to j}$ denotes the interaction term for the $p$-th channel (hence a scalar value).
>
> In addition to correcting this, we will:
> - remove duplicated references,
> - add appropriate citations (Sinkhorn, MFM, SF2M),
> - and fix minor typographical errors throughout.
>
> ---
>
> Thank you again for the detailed feedback. We hope our responses have fully addressed your comments, and we would be glad to clarify any remaining points.

---

> > ### Author Rebuttal · Reviewer_1LJW · 2026-04-05
> >
> > The authors have addressed my questions.

---

> > > ### Author Response · Authors · 2026-04-07
> > >
> > > Thank you for your response, we are glad that your questions have been fully resolved, and want to thank you for your constructive feedback in this review process!

---

### Decision · Program_Chairs · 2026-04-30

**Decision:**

Accept (regular)

**Comment:**

The paper suggests an optimal transport approach for aligning single-cell RNA-seq data across multiple time snapshots, that includes a Gromov-Wasserstein term to match cell-cell affinities computed based on known ligand--receptor gene pairs. The authors show that including this GW term leads to performance improvements on held-out data.

The reviewers noticed that the numerical improvement was modest, but nevertheless liked the overall approach and presentation clarity. I agree. I think the paper is well done, and I liked the way the experiments were set up, including ablation experiments. I recommend acceptance.

Some further comments from my side:

* Figure 1 is very unclear. At this point the reader does not know what G is. What do the rows correspond to, is unclear. The caption does not explain what alpha is, and what alpha-star is. Even after reading the paper, I am not sure I interpret the figure correctly. Does D_0 have 4 cells or 8 cells here? I guess 8, otherwise I don't understand the arrows on the right. But then why do the upper 4 cells have exactly the same constellation as the lower 4 cells? That's very confusing.

* Section 3, "feature-based priors" -- you write that these approaches operate within a single snapshot. Why? It is common to pool all snapshots together and then use kNN graph to infer the pseudotime etc. Your formulation is confusing to me.

* Section 3, "dynamic priors" -- new paragraph

* Section 4.1, define library-size normalization

* Section 4.1, using Hill saturation instead of log-transform is very unusual, and the motivation is unclear. Did you try various approaches here? When you are comparing your method to other methods, do you always preprocess the data using Hill transform? What is the value of h_g?

* Section 4.1, optimization: are you not using entropic regularization? This is unusual, can you discuss this?

* I was super confused by Figure 3. I understand that each cluster is translated by a separate vector. But what does this have to do with G_ij interactions? What interactions are used here for the GW term? I don't understand it either from the figure or from the text. Please make sure to explain this properly.

* Figure 4: the font is way too small.

* Section 5.1: why does Dendritic Stimulus dataset behave differently? Can you comment?

* Table 1 does not show alpha=0 rows for CellBRIDGE, even though the text says that it does!!

* Table 2 values for CellBRIDGE do not match to the values in Table 1, why is that? This is very confusing.

* Figure 5 caption: "limited benefit" -> "no benefit".